# DDA: Dimensionality Driven Augmentation Search for Contrastive Learning in Laparoscopic Surgery

**Yuning Zhou[1] Henry Badgery[2] Matthew Read[3] James Bailey[4]**
**Catherine E. Davey[1]**

[1] *Department of Biomedical Engineering, the University of Melbourne, Australia*

[2] *Department of HPB/UGI Surgery, St Vincent's Hospital Melbourne, Australia*

[3] *Department of Surgery, St Vincent's Hospital Melbourne, Australia*

[4] *School of Computing and Information Systems, the University of Melbourne, Australia*

**Editors:** Accepted for publication at MIDL 2024

## Abstract

Self-supervised learning (SSL) has potential for effective representation learning in medical imaging, but the choice of data augmentation is critical and domain-specific. It remains uncertain if general augmentation policies suit surgical applications. In this work, we automate the search for suitable augmentation policies through a new method called Dimensionality Driven Augmentation Search (DDA). DDA leverages the local dimensionality of deep representations as a proxy target, and differentiably searches for suitable data augmentation policies in contrastive learning. We demonstrate the effectiveness and efficiency of DDA in navigating a large search space and successfully identifying an appropriate data augmentation policy for laparoscopic surgery. We systematically evaluate DDA across three laparoscopic image classification and segmentation tasks, where it significantly improves over existing baselines. Furthermore, DDA's optimised set of augmentations provides insight into domain-specific dependencies when applying contrastive learning in medical applications. For example, while hue is an effective augmentation for natural images, it is not advantageous for laparoscopic images.

**Keywords:** differentiable augmentation search, contrastive learning, laparoscopic imaging

## 1. Introduction

Self-supervised learning (SSL) has recently shown its potential for generating representations from large-scale datasets without human supervision (Chen et al., 2020a; Grill et al., 2020; Bardes et al., 2022; He et al., 2022). In this approach, a model typically conducts representation learning using a data-generated objective on unlabeled datasets. The learned representations can subsequently be transferred to downstream tasks with limited annotations. In medical domains such as endoscopic or laparoscopic surgery, data can be readily generated from surgical recordings. This contrasts with obtaining high-quality annotations, which can be prohibitively expensive in terms of human expert time, particularly for applications like segmentation (Ward et al., 2021). In such cases, SSL is highly attractive as it allows effective utilisation of unlabeled data, thereby reducing the demand for annotations.

Contrastive learning is a type of SSL, where the model minimizes the distance between feature representations of augmented views from the same image and maximizes the distance to different images. This can be achieved either explicitly (Chen et al., 2020a) in the loss

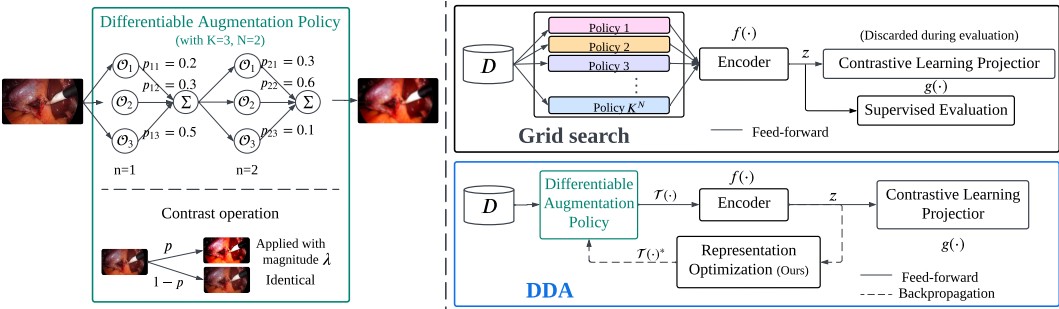

Figure 1: Illustration of differentiable augmentation policy design and an example application of the contrast operation (left), and the comparison of grid search and our DDA framework for contrastive learning (right).

function or implicitly (Grill et al., 2020; Bardes et al., 2022). The augmentation policy that generates augmented views significantly influences the representation learning quality, and consequently the transferred performance on downstream tasks (Wagner et al., 2022; Huang et al., 2023). Existing augmentation search methods that automatically select the "optimal" policy are usually conducted with supervised learning and require access to annotations (Cubuk et al., 2019; Lim et al., 2019; Ho et al., 2019; Hataya et al., 2020). Reed et al. (2021) proposed SelfAugment for augmentation policy search in contrastive learning. It requires training additional projectors with proxy SSL tasks for each candidate and uses Bayesian optimisation for selection. It is very time-consuming if the search space is large.

Given that the most effective augmentation policy is domain dependent (Xiao et al., 2021; Bendidi et al., 2023), the transferability of policies that have been developed for natural images (Deng et al., 2009; Chen et al., 2020a; Grill et al., 2020; Garrido et al., 2023) to medical datasets remains uncertain. Van der Sluijs et al. (2023) explored contrastive learning augmentation policies on Chest X-ray datasets by conducting a grid search on a predefined space of 8 augmentations with varying strengths. The $8^2$ search space poses challenges for replication due to the exponential growth in computational resources required for SSL (Chen et al., 2020b). This challenge is further pronounced when extending findings to other medical applications.

To address these limitations, we propose *Dimensionality Driven Augmentation (DDA)* as illustrated in Figure 1. to streamline augmentation policy selection in contrastive learning. Based on the differentiable augmentation search framework (Hataya et al., 2020), DDA incorporates the augmentation pool, the number of augmentations in a policy, their probabilities and strengths of application as differentiable parameters. Instead of optimizing performance on downstream tasks, which is often infeasible, we optimise a proxy objective function. DDA considers the optimisation of geometric characteristics of the deep representation in contrastive learning (the intrinsic dimension). As a result, DDA does not require access to an annotated dataset or additional training. We demonstrate that DDA can explore a large search space, encompassing up to $10^8$ different choices, with an optimizable augmentation strength.

To summarize, the contributions of this paper are:

- We propose a novel approach, DDA, to search for optimal augmentation policy in contrastive learning without any additional supervised evaluation (finetuning) on an-

notated datasets. DDA effectively identifies a suitable augmentation policy for laparoscopic surgery across various tasks.

- Notably, DDA significantly reduces the time required for augmentation search to constant time complexity. For example, navigating a $10^5$ search space with DDA only takes 48 hours, which is $10^5$ times faster compared to a grid search.

- We show that commonly used augmentations for natural images are sub-optimal when used for laparoscopic images. Our selected augmentations provide valuable insights into which techniques are most effective for medical applications.

## 2. Method

In this section, we first provide background regarding contrastive learning and its augmentation policy in Section 2.1. Then, we describe the learnable augmentation policy design and an overview of the DDA framework in Section 2.2, followed by a detailed discussion on our proxy objective function in Section 2.3.

### 2.1. Problem Definition

**Contrastive Learning.** We consider a typical two-stage semi-supervised setting, a popular SSL application. We are given a dataset, $\mathcal{D}$, comprised of an unlabelled subset $\mathcal{D}_u$ for contrastive pretraining, and a labelled subset $\mathcal{D}_l$ for finetuning, where $\mathcal{D} = \mathcal{D}_u \cup \mathcal{D}_l$.

In the first stage, for each $\bar{\boldsymbol{x}} \in \mathcal{X}$ in a batch of $M$ images from $\mathcal{D}_u$, we generate two positive samples $\boldsymbol{x}, \boldsymbol{x}^+$ from the augmentation policy $\mathcal{T}(\bar{\boldsymbol{x}})$, and $2(M-1)$ independent negative samples $\{\boldsymbol{x}_m^-\}_{m=1}^{2(M-1)} = \{\mathcal{T}(\boldsymbol{x})|\boldsymbol{x} \neq \bar{\boldsymbol{x}}\}$ from the other $M-1$ images. An encoder $f(\cdot)$ maps input images to the representation space $\mathcal{X} \to \mathcal{R}^d$ with the representation $\mathbf{z} = f(\boldsymbol{x})$, and projector $g(\cdot)$ obtains embedding $\mathbf{e} = g(\mathbf{z}) \in \mathcal{R}^e$. A classical optimisation objective for contrastive learning (Chen et al., 2020a) is the following,

$$\mathcal{L}_{\text{NTXent}} = \mathbb{E}_{\boldsymbol{x},\boldsymbol{x}^+,\{\boldsymbol{x}_m^-\}} - \ln \frac{\exp(\text{sim}(\boldsymbol{e}, \boldsymbol{e}^+)/\eta)}{\sum_{m=1}^{M} \exp(\text{sim}(\boldsymbol{e}, \boldsymbol{e_m})/\eta)}, \tag{1}$$

where $\text{sim}(\cdot)$ is the cosine similarity, and $\eta > 0$ is the temperature that controls the smoothness of distance distribution.

In the second stage, after the encoder is trained, an additional classifier or segmentation model $h(\cdot)$ can be attached to the encoder as $h \circ f$, for simplicity, we denote this as $f'$. The $f'$ learns to map the input image to the label space $\mathcal{X} \to \mathcal{Y}$ using $\{(\boldsymbol{x}, \boldsymbol{y})\} \in \mathcal{D}_l$ following a typical supervised learning setup by minimizing the following objective,

$$\mathcal{L}_{\text{Supervised}} = \mathbb{E}_{\boldsymbol{x} \sim \mathcal{D}_l} \mathcal{L}(f'(\boldsymbol{x}), \boldsymbol{y}), \tag{2}$$

where $\mathcal{L}$ is the objective function for supervised downstream tasks (e.g., cross-entropy).
**Augmentation Policy.** The optimal policy for contrastive learning is defined as,

$$\mathcal{T}^* = \arg\min_{\mathcal{T}} \mathbb{E}_{\boldsymbol{x} \sim \mathcal{D}_{Val}} \mathcal{L}(f'(\boldsymbol{x}), \boldsymbol{y}), \tag{3}$$

where $\mathcal{D}_{Val}$ is the unseen validation data. Note that the above objective is defined with respect to a specific downstream task. However, in contrastive learning, we are really interested in obtaining the optimal $\mathcal{T}^*$ that is suitable for a wide range of tasks.

## 2.2. DDA Search Framework

Each round of a two-stage learning framework mentioned in the previous section is time consuming. This challenge makes it impossible to apply a thorough grid search, especially for a large search space with a diverse number of operations and choices in each augmentation policy. To tackle these challenges, we propose a streamlined search framework, DDA.

Specifically, we setup the augmentation search in a differentiable fashion following existing works (Liu et al., 2019; Hataya et al., 2020). Let $\mathbb{O}$ be a set of $K$ image augmentation operations $\mathcal{O}_k \in \mathbb{O} : \mathcal{X} \to \mathcal{X}$, and $\mathcal{X}$ is input space. Each operation $\mathcal{O}_k(\,\cdot\,;p_k,\lambda_k)$ has two parameters: the probability $p_k$, and the augmentation magnitude $\lambda_k$, for applying the operation. For an input image, $\boldsymbol{x} \in \mathcal{X}$, the output of applying an augmentation from the set of possible operations $\mathbb{O}$, also known as a sub-policy, depends on weighted sampling. This is defined as, $\bar{\mathcal{O}}(\boldsymbol{x};\boldsymbol{p},\boldsymbol{\lambda}) \in \{\mathcal{O}_k(\boldsymbol{x};p_k,\lambda_k)|k=1,\dots K\}$.

Let $\tau$ be the set of augmentation sub-policies as $\tau = \{\bar{\mathcal{O}}_1,\dots\bar{\mathcal{O}}_{N_\tau}\}$ contains $N_\tau$ consecutive sub-policies. The augmented image is obtained by $\boldsymbol{x}' = \tau(\boldsymbol{x}) = (\bar{\mathcal{O}}_{N_\tau} \circ \cdots \circ \bar{\mathcal{O}}_1)(\boldsymbol{x};\boldsymbol{p_n},\boldsymbol{\lambda_n})$, $n = 1,\dots N_\tau$. During searching, the output of the $n^{th}$ sub-policy $\bar{\mathcal{O}}_n(\boldsymbol{x},\boldsymbol{p_n},\boldsymbol{\lambda_n})$ is the weighted sum of all possible operation choices,

$$\bar{\mathcal{O}}_n(\boldsymbol{x},\boldsymbol{p_n},\boldsymbol{\lambda_n}) = \sum_{k=1}^{K} \mathcal{O}_k(\boldsymbol{x};\sigma_\eta(w_{nk}),\lambda_{nk}), \tag{4}$$

The probability $p_k$ is convert from learnable real-number parameter, $w_k$, using the softmax function as $p_{nk} = \sigma_\eta(w_{nk}) = \frac{\exp(w_{nk}/\eta)}{\sum_K \exp(w_{nk}/\eta)}$, with temperature $\eta > 0$. Low temperature will generate the distribution of $p_k$ as a one-hot like vector.

The components of the augmentation policy thus become learnable parameters and can be updated by backpropagation. In this case, our augmentation policy is designed as a block of plug-in-and-play operation layers as $\mathcal{T}$ with parameters $\boldsymbol{w},\boldsymbol{\lambda}$. To learn the optimal augmentation policy $\mathcal{T}^*$, we further propose a suitable proxy objective in Section 2.3 for the augmentation search, which optimises the contrastive representation on a fixed encoder.

Here we provide an overview of the three-step DDA search framework as illustrated in Figure 1. Firstly, we obtain a contrastive learning encoder $f$ that is trained with the most basic augmentation (e.g. image cropping) with a contrastive projector $g$. Then, we apply our differentiable augmentation search with the proxy objective explained in Section 2.3 to optimise the parameters of the policy only on the fixed $f$. Lastly, the optimal policy $\mathcal{T}^*$ will be applied to re-conduct contrastive learning to obtain $g^* \circ f^*$. We provide a sketch of the pseudocode of the DDA framework in Algorithm 1. in Appendix A.

## 2.3. DDA Search with Representation Dimensionality

Equation (3) can't be directly optimised due to lacking validation data, and the two-stage training is hard to optimise differentially. We propose a novel objective as a proxy so that the differentiable search only involves contrastive pretraining.

In contrastive learning, the deep representation $\mathbf{z} \in \mathcal{R}^d$ extracted from a surgical image needs to be distinguishable from the representations of other images. Our focus lies in understanding the local distribution of deep representations in vicinity of a query image. We define a local neighbourhood-of-interest as a $d$-dimensional sphere with a small radius

$r$ centered at the query. Any data within the neighbourhood has a smaller distance than $r$ from the query, and likely has similar visual content as the query. An illustration of the deep representation for a local neighbourhood is shown in Figure 4, in Appendix A.

To evaluate the properties of a query image's neighborhood, one can estimate the growth rate in the number of nearby data points encountered as the distance from the query increases. This growth rate provides an estimation of the local intrinsic dimensionality (LID) of the query data located in the high-dimensional subspace (Houle, 2017). Intuitively, the LID assesses the effective number of dimensions (the intrinsic dimension) needed to characterize the local neighbourhood of a query point. If the LID equals 2, the local neighbourhood surrounding the query point behaves like it has two dimensions. LID is thus like a complexity measure and assesses the space-filling properties around a query point. In our scenario, we will assess the LID of each point in the deep representation produced by SSL.

Let $\mathbf{z}$ be the query point, and $\mathbf{s}$ denotes any nearby representation of a different image within its vicinity. The distance between $\mathbf{z}$ and $\mathbf{s}$ is denoted as $r_s = l(\mathbf{z}, \mathbf{s})$, where $l$ represents a distance measurement. We define the cumulative distribution function (CDF), $F(r)$, of the number of data points concerning the local distance distribution with respect to $\mathbf{z}$ as the probability of the sample distance lying within a threshold $r$, $F(r) \triangleq \Pr[r_s \leq r]$.

**Theorem 1 (Houle (2017))** *If $F$ is continuously differentiable at $r$, then*

$$\mathrm{LID}_F(r) \triangleq \frac{r \cdot F'(r)}{F(r)} \,.$$

The local intrinsic dimension (LID) at $\mathbf{z}$ defined as the limit, when the radius $r$ tends to zero as,

$$\mathrm{LID}_F^* \triangleq \lim_{r \to 0^+} \mathrm{LID}_F(r) \,.$$

In practice, $\mathrm{LID}_F^*$ needs to be estimated, and it is not an integer. For simplicity, for the rest of the paper, we denote 'LID' as the quantity of $\mathrm{LID}_F^*$.

In our scenario, we will wish to optimise the LID of each data sample, encouraging it to be large. Work by Huang et al. (2024) has shown that it is theoretically desirable to optimise the log transform of the LID in deep learning, rather than the LID. Intuitively, if a query point has a large (log) LID, then its neighbourhood requires more dimensions to characterise. This is a desirable property for SSL, where a key objective is to avoid what is known as dimensional collapse (Jing et al., 2022).

This makes LID a suitable proxy objective for the augmentation search. Since 1) it does not require finetuning on downstream tasks, and 2) optimising the LID distribution of samples in the deep space can easily be integrated into a gradient descent framework. Following Huang et al. (2024), to obtain a representation with higher LIDs, we can optimise the Fisher-Rao distance between the LID of the representations $\boldsymbol{z}$ and a uniform distance distribution (with LID of 1) using the following loss function as,

$$\mathcal{L}_{\mathrm{DDA}} = \min -\frac{1}{M} \sum_i^M \ln \mathrm{LID}_{F_i}^* \,, \tag{5}$$

where M is the number of samples in the batch, and $\text{LID}^*_{F_i}$ can be estimated using any popular estimator for LID in the encoder's output representation $z$.

Different from existing work (Huang et al., 2024) that investigated optimisation of model parameters to produce a better contrastive representation, *in our work we optimise the augmentation policy with a fixed encoder $f$*. We find this simple but important shift in perspective is highly significant. More concretely, we apply differentiable augmentation search to optimise the parameters of the policy with Equation (5). as the objective as shown in Algorithm 2 in the Appendix A. Although it is also possible to apply other proxy functions instead of using LID for DDA, we empirically find that using LID is highly suitable for the augmentation search. *To our knowledge, ours is the first work to consider the use of LID as a measure for optimising augmentations.*

## 3. Experiments

We evaluate the performance of the augmentation policy found by DDA in terms of representation quality following standard evaluation protocols such as linear evaluations and finetuning. We use ResNet-50 (He et al., 2016) and SimCLR (Chen et al., 2020a) as the contrastive learning framework. All hyperparameters closely follow the original papers. We perform the pretraining on two datasets: our private dataset SVHM and the public dataset Cholec80 (Twinanda et al., 2016). For downstream evaluations, we use the Cholec80 Tool (Twinanda et al., 2016), CholecSeg8K (Hong et al., 2020) and our annotated private dataset (denoted as SVHM Seg). For the augmentation search, we set $N = 5$ by default. We search across the following augmentation operations: *Identical, Brightness, Contrast, Hue, Saturation, Solarize, GaussianBlur, Posterize, Gray, Sharpness*, with further details in Appendix B.2. We set the temperature $\eta$ in Equation (4). to 0.1. For LID estimation, we use the method of moments estimator (Amsaleg et al., 2018), with the neighbourhood size 16. We compare with the original augmentation policy used by SimCLR, a *Base* policy (only with crop and horizontal flip), a randomly generated one using our search space (*Random*), a manually selected one based on domain experts (*Manual*), and SelfAugment (Reed et al., 2021) with differentiable adaptations. For SelfAugment, we use its min-max strategy as it showed the best performance. We use categorical sampling of the final policy as default. Additional results for the Argmax sampling can be found in Appendix B.4. A description of our private dataset, technical details, computing infrastructure and sample source code is provided in Appendix B. The augmentation search using DDA takes 8 hours for our private dataset and 3 hours for Cholec80.

### 3.1. Evaluations

As shown in Table 1, compared against the original SimCLR augmentation, a large 6% improvement can be observed for our DDA on the linear evaluations. The original SimCLR policy performs similarly to the Base policy (crop and resize) on laparoscopic cholecystectomy datasets. For finetuning on segmentation tasks, augmentations found by our method can outperform the original SimCLR policy by 1-3%. This can be considered as a significant improvement in the context of SSL evaluation (He et al., 2020; Bardes et al., 2022). Without the augmentation search, using randomly selected operations with our search space results in a complete collapse of the representations, where the model outputs a constant

Table 1: All results are based on using ResNet-50 as encoder and SimCLR as contrastive pretraining. Results of the linear probing are reported using mean Average Precision (mAP) (%), and finetuning on downstream segmentation tasks is reported using mIoU (%). The best results are in **boldface**.

| Pretraining Dataset | Augmentation | Sampling | Linear Prob (Classification) | Finetune (Segmentation) | |
| --- | --- | --- | --- | --- | --- |
| | | | Cholec80 Tool | SVHM Seg | CholecSeg8K |
| None | Supervised | - | - | 55.77 | 57.79 |
| SVHM | SimCLR | N/A | 60.00 | 57.28 | 56.18 |
| | Base | N/A | 59.01 | 57.15 | **58.71** |
| | Manual | N/A | 47.97 | 54.09 | 57.11 |
| | Random | Categorical | Complete Collapse | N/A | N/A |
| | SelfAugment | Categorical | Complete Collapse | N/A | N/A |
| | DDA | Categorical | **65.95** | **58.29** | 57.86 |
| Cholec80 | SimCLR | N/A | 67.59 | 58.29 | 56.02 |
| | Base | N/A | 67.78 | 58.36 | 57.04 |
| | Manual | N/A | 59.00 | 55.12 | 59.01 |
| | Random | Categorical | Complete Collapse | N/A | N/A |
| | SelfAugment | Categorical | 60.24 | 58.41 | 55.86 |
| | DDA | Categorical | **73.59** | **59.31** | **59.40** |

vector (Jing et al., 2022). It has been shown in existing work (Jing et al., 2022) that excessive augmentations could cause dimension or complete collapse. It is worth noting that the augmentation policy found by DDA avoids selecting an excessive number of operations by selecting *Identical*, as shown in Figure 2(c) and 2(d). Compared with SelfAugment, our method also demonstrated consistently superior performance. On our private SVHM dataset, the augmentation policy found by SelfAugment caused a complete collpase. For pretraining with Cholec80, DDA outperforms SelfAugment by 13% in the linear evaluations.

### 3.2. Analysis of the Augmentation Policy Found by DDA

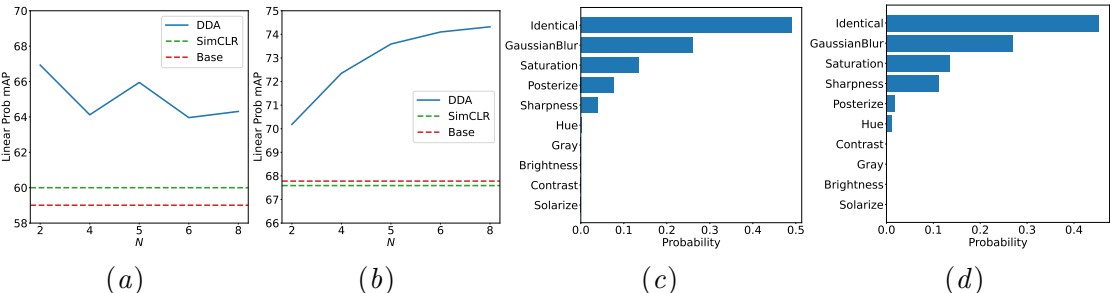

$(a)$ $(b)$ $(c)$ $(d)$

Figure 2: (a-b) Linear probing accuracy on the Cholec80 Tool dataset with different numbers of augmentation operations ($N$). Each data point is an individual run of the experiment, from augmentation search to pretraining and evaluations. (c-d) Distributions of different augmentation operations found by our method. In subfigures (a) and (c), results are obtained by pretraining on our private SVHM dataset. In subfigures (b) and (d), results are obtained by pretraining on the public dataset Cholec80.

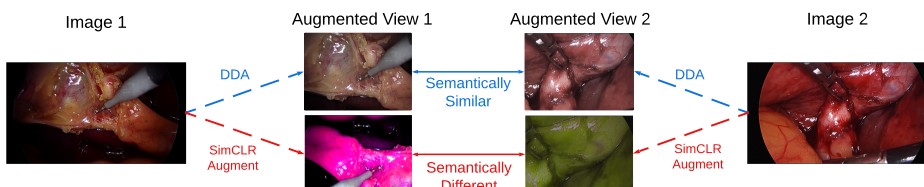

Figure 3: Illustration of DDA and SimCLR augmented images on SVHM dataset.

In this subsection, we investigate the effect of the different numbers of operations in the policy for the representation quality. We also examine the policy found by DDA. As shown in Figure 2(a) and 2(b), the representation quality evaluated by the linear probing shows a consistent improvement over the augmentation used by SimCLR and the Base policy. Similar results for finetuning can be found in Appendix B.4. This consistent improvement indicates that the commonly used augmentations on natural images appear suboptimal for a medical dataset such as laparoscopic cholecystectomy (LC).

To further investigate the difference between the optimal policy for natural images and policies found by DDA, which are suitable for the medical datasets, we plotted the distributions of selected operations in Figure 2(c) and 2(d). It can be observed that despite the different pretraining datasets (our private dataset SVHM and Cholec80), the resulting policy is similar. Detailed results for each policy are in Appendix B.3. The policy found by DDA prefers selecting *Gaussian Blur*, *Saturation*, *Posterize*, and *Sharpness*. One common characteristic regarding these operations is that they do not change the colour profile of the image. In the existing literature, it has been shown that an augmentation resulting in an overlapping view of two different images and semantically similar to each other is beneficial for contrastive learning (Cai et al., 2020; Wang et al., 2022; Huang et al., 2023; Joshi and Mirzasoleiman, 2023). In the context of LC images, randomly changing the colour profile to other random colours is not ideal for creating such an overlapping view, since the majority of the contents in the images are red. On the other hand, the *Gaussian Blur* and *Posterize* could create a blurring effect or decrease the image quality of the image that could easily create an overlapping view. This is because in LC surgery, motion movement of the camera and fog generated during anatomy dissection by diathermy hook can degrade image quality. Similarly, the effect of *Sharpness* and *Saturation* could also inherently appear in the dataset. In Figure 3, we provide a visualisation of augmented images using DDA and SimCLR augmentation, where DDA can create semantically similar views while SimCLR augmentation produces different and unrealistic views. Additional augmented images can be found in Appendix C. In summary, the augmentation policy found by DDA is more effective (verified by linear probing and finetuning) and well suited to the LC dataset.

## 4. Conclusion

In this study we introduced DDA, an automatic augmentation search method explicitly tailored for contrastive learning which considers dimensionality characteristics of the deep representation. DDA showcases both efficiency and effectiveness in identifying inherently optimised augmentation policies for laparoscopic images. Beyond its application to laparoscopic images, DDA has the potential to be used in other contexts, particularly in representation learning for medical images.

## Acknowledgments

This research was supported by The University of Melbourne's Research Computing Services and the Petascale Campus Initiative. All data are provided with ethics approval through St Vincent's Hospital (ref HREC/67934/SVHM-2020-235987).

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

## Appendix A. DAA Algorithm

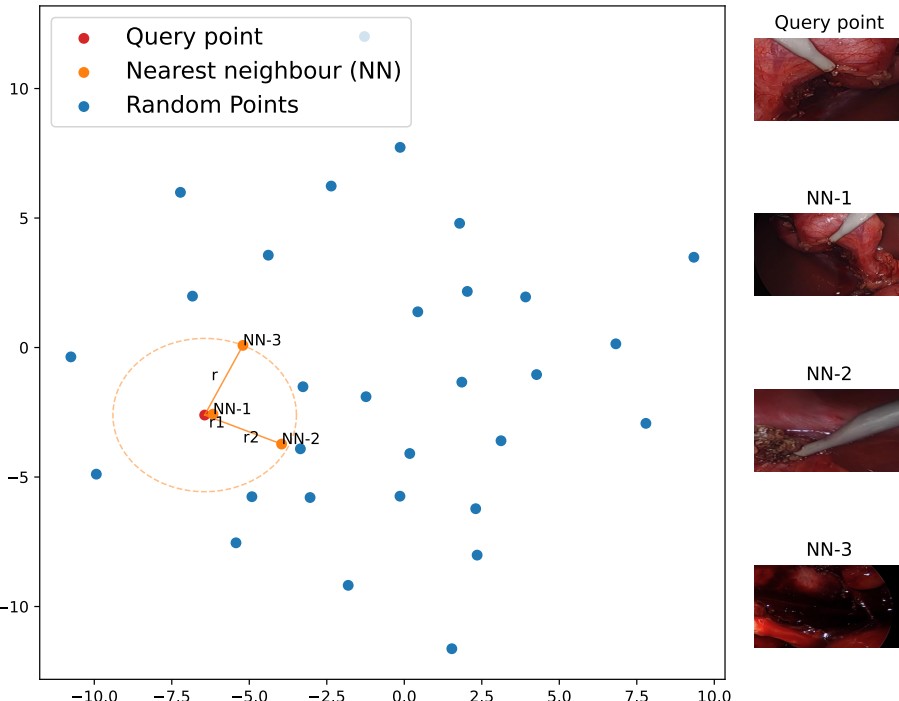

Figure 4: The figure shown on the left is a representation learned by training a SimCLR model with DDA. These representations are projected onto a 2D space using t-SNE. In this visualization, the radius, $r$, indicates the maximum distance from the query to the relevant neighbourhood, while $r1$ and $r2$ represent the distances from the query to the first (NN-1) and second (NN-2) nearest data points, respectively. The third nearest neighbour (NN-3) lies on the sphere at the same distance from the query as $r$.

---

**Algorithm 1:** Using DDA to obtain a pre-trained model (higher level pipeline)

---

**Input:** Encoder $f$, projector $g$, unlabeled dataset $\mathcal{D}_u$, basic augmentation $\mathcal{T}_{basic}$, augmentation policy $\mathcal{T}_{\boldsymbol{w},\lambda}$

**Output:** Optimal policy $\mathcal{T}^*_{\boldsymbol{w}^*,\lambda^*}$, final encoder pretrained with optimal policy $f^*$

Step 1: Conduct contrastive learning for $g \circ f$ on $\mathcal{D}_u$ using $\mathcal{T}_{basic}$

Step 2: Optimise $\mathcal{T}_{\boldsymbol{w},\lambda}$ on fixed pre-trained $f$ to find $\mathcal{T}^*_{\boldsymbol{w}^*,\lambda^*}$ through DDA search (see Algorithm 2)

Step 3: Redo contrastive learning for randomly initialised model $g \circ f$ on $\mathcal{D}_u$ using $\mathcal{T}^*_{\boldsymbol{w}^*,\lambda^*}$ to obtain final $f^*$

---

In Figure 4, we provide an intuitive example of LID, which describes the relative rate at which its cumulative distance function (CDF), the $F(r)$ increases as the distance $r$ increases from 0. We use the representation learned by SimCLR and project it into a 2D space using t-SNE as an example. Considering a radius $r$ for the query point (the red dot), the LID

---

**Algorithm 2:** DDA Search

---

**Input:** Encoder $f(\cdot)$, Dataset $\mathcal{D}_u$, policy $\mathcal{T}_{\boldsymbol{w},\lambda}$, learning rate $\alpha$, neighbourhood size $k$, number of epochs $E$, basic augmentation $\mathcal{T}_{basic}$

**Output:** Optimal policy $\mathcal{T}^*_{\boldsymbol{w}^*,\lambda^*}$

Conduct contrastive learning for $f(\cdot)$ on $\mathcal{D}_u$ with $\mathcal{T}_{basic}$

**for** $e \leftarrow 1$ **to** $E$ **do**

    **for** $i \leftarrow 1$ **to** *Number of Batches* **do**

        $\boldsymbol{x} = \text{Sample}(\mathcal{D}_u)$ ;                    /* Random sample batch of images */

        $\boldsymbol{x}' = \mathcal{T}_{\boldsymbol{w},\lambda}(\boldsymbol{x})$ ;                        /* Augment images */

        $\mathbf{z} = f(\boldsymbol{x}')$ ;                       /* Obtain representations */

        $\text{LID}_{\boldsymbol{x}'} = \text{estimations}(\mathbf{z}, k)$ ;          /* LID estimations */

        $\mathcal{L} = - \log(\text{LID}_{\boldsymbol{x}'})$ ;            /* Follow Equation (5) */

        $(\boldsymbol{w},\lambda)^{i+1} = (\boldsymbol{w},\lambda)^i - \alpha\nabla\mathcal{L}((\boldsymbol{w},\lambda)^{(i)})$ ;   /* Gradient descent on $\mathcal{T}$ parameters */

    **end**

**end**

---

measures the rate of growth in the number of data objects (blue dots) encountered as the radius $r$ increases. The LID can be estimated by calculating the distances to its k-nearest neighbourhoods (orange dots), such as Maximum Likelihood Estimator (MLE) (Levina and Bickel, 2004) and the Method of Moments (MoM) (Amsaleg et al., 2018).

In Algorithm 2, we present the pseudo-code for applying DDA. On a high level, we use basic augmentation to train a contrastive learning encoder and then use this encoder to obtain representations of the data. We optimize a differentiable augmentation policy $\mathcal{T}^*_{\boldsymbol{w}^*,\lambda^*}$ with randomly initialized parameters that can maximize the LID of the representations. The optimized policy can then be used to train the final encoder with contrastive learning.

## Appendix B. Experiments

We conducted our experiments on Nvidia A100 GPUs with PyTorch implementation, with each experiment distributed across 4 GPUs. We used automatic mixed precision due to its memory efficiency. The implementation of the differentiable search following existing work (Hataya et al., 2020) and their official code base, as well as the Kornia library[1]. Our implementation is available in this code repository: https://github.com/JoJoNing25/DDAug.

### B.1. Experiment Settings

In this section, we provide details regarding our experiment settings.

**Pretraining.** For contrastive pretraining, we use SimCLR (Chen et al., 2020a) and ResNet-50 (He et al., 2016), pretraining for 50 epochs, with a base learning rate of 0.075 with the square root scaling rule. We use LARS (You et al., 2017) as the optimiser and weight decay of $1 \times 10^{-6}$. We use batch size of 192 for all experiments.

We constructed a large-scale in-house dataset (SVHM) by prospectively recording 70 laparoscopic cholecystectomy (LC) videos, each for an operative case with diverse disease

---

1. https://github.com/kornia/kornia

severity and anatomical variability (Madni et al., 2018), over three major teaching hospitals in Australia, the Epworth HealthCare, St Vincent's Hospital Melbourne public and private sectors. Procedures followed the same supine approach (Carroll, 1995), and we only considered data from the initial grasping of the gallbladder up to the cutting of the cystic duct in $1,960 \times 1,080$ pixels.

We perform the pretraining with public datasets Cholec80 (Twinanda et al., 2016), and our private SVHM dataset. Cholec80 has 400,000 frames from 80 videos at 2 fps. We use 70 videos for the pretraining and resevering the rest 10 vidoes for downstream evaluations. For our private SVHM dataset, we use 300,000 images dissected at 4 frame-per-second (fps) from 50 videos. We use image size of $480 \times 854$ pixels for pretraining with Cholec80 and $432 \times 784$ for our private dataset.

**Linear Probing and Finetuning.** For downstream evaluations, we use the Cholec80 tool classification dataset (Twinanda et al., 2016) for linear probing, CholecSeg8k (Hong et al., 2020) and our private labelled dataset (SVHM Seg).

Cholec80 tool dataset (Twinanda et al., 2016) is a multi-class classification task to determine if the sugical tool is presence in the image. We use 70 videos for training and the rest 10 videos for evaluations. There is no overlap between the videos for evaluations and training/pretraining.

CholecSeg8k (Hong et al., 2020) is a labeled subset of Cholec80 where 8,080 frames of $480 \times 854$ at 25 fps are extracted from 17 videos in Cholec80. We removed frames from training set that are presence in the test set to ensure no leakage.

Our SVHM Seg dataset is collected from 20 videos and yielded 1,975 frames with a training dataset of 1,583 frames from 16 videos and a test set of 392 frames from 4 videos unseen in the training set. The dataset was annotated with 20 classes and validated by our surgeons, details are in Table 2. There is no overlap between the SVHM Seg test set, the SVHM Seg training set, or the pretraining SVHM dataset since they are split based on videos (distinctive operating case).

Table 2: Class description of our SVHM Seg dataset.

| Class name | Description |
| --- | --- |
| Abdominal wall | abdominal wall |
| Background | black background beyond circular visual field |
| Cholangiogram catheter | instrument to apply dye-enhanced imaging for bile ducts visulization (includes shaft, trip and catheter) |
| Clip applicator | instrument to apply clips to close cystic artery and duct (includes shaft, trip and catheter) |
| Common bile duct | bile duct drain from hepatic ducts to duodenum |
| Cystic artery | blood supply to the gallbladder |
| Cystic duct | duct draining bile from gallbladder to common bile duct |
| Diathermy hook shaft | diathermy hook instrument - shaft |
| Diathermy hook tip | diathermy hook instrument - tip |
| Duodenum | dection of gastrointestinal tract where common bile duct drains, distal to stomach |
| Gallbladder | gallbladder |
| Grasper shaft | grasping instrument of any kind - shaft |
| Grasper tip | grasping instrument of any kind - tip |
| Liver | all other liver segments |
| Omentum | intra-abdominal fat, includes small bowel |
| Rouviere's sulcus | cleft on the right side of the liver; important landmark |
| Scissors shaft | instrument to cut tissues and structures |
| Scissors tip | instrument to cut tissues and structures |
| Segment iv | segment of liver to the patient left side of gallbladder |
| Sucker irrigator | cylindrical instrument for suction and irrigation |

For linear probing, we follow the standard protocol that adds a linear classification layer on top of the frozen encoder. Following Ramesh et al. (2023), for the Cholec80 tool dataset, we use weighted binary cross entropy loss. We use SGD as an optimiser with a learning rate of 0.1, weight decay of $1.0 \times 10^{-4}$, and batch size of 256 and 80 epochs.

For finetuning on the segmentation task, DeepLabV3+ (Chen et al., 2017) is used for the segmentation head. For all datasets, we use AdamW (Loshchilov and Hutter, 2019) as the optimiser with a learning rate of 0.005, weight decay of 0.05, and batch size of 32 and 100 epochs. We use the same image size as used in pretraining.

## B.2. Search Space of the Augmentation

We summarized all selected operations for the search space in Table 3. The augmentation policy used by SimCLR (Chen et al., 2020a) on ImageNet (Deng et al., 2009) is summarized in Table 4. For easy comparison, we converted the strength used by SimCLR into the same scale as our search space. We perform the search for 10 epochs with a learning rate of 0.01, and Adam (Kingma and Ba, 2014) as the optimiser. This takes around 8 hours for our private dataset and 3 hours for Cholec80.

For comparison with the *Manual*, it is manually selected augmentation based on domain expert. In supervised learning, existing works (Tokuyasu et al.; Silva et al., 2022; Scheikl et al., 2020; Owen et al., 2022) have shown that rotation by 30 degrees, contrast, Gaussian noise, and Gaussian blur are commonly used for supervised segmentation tasks. Based on this domain knowledge, we constructed an additional manual selection policy using these popular augmentations from the literature. We set the probability of applying each augmentation to 0.8. The strength for rotation is 30 degrees, the strength for contrast and Gaussian noise is randomly sampled, and the sigma is set to 0.1 to 2.0 for Gaussian blur based on the settings from the above-mentioned literature.

For comparison with the baseline method, the SelfAugment (Reed et al., 2021), for fair comparison and efficiency, we use our differentiable framework instead of the original Bayesian optimisation. This is more efficient because only one additional proxy linear layer is required. We adopt the original proxy SSL task, rotation for the proxy linear layer, and Min-Max (minimize $\mathcal{L}_{SS}$ and maximize $\mathcal{L}_{NTXent}$) loss objective defined as the following:

$$\underset{\mathcal{T}}{\arg\min}\, \mathcal{L}_{SS} - \mathcal{L}_{NTXent}, \tag{6}$$

where the $\mathcal{L}_{SS}$ is the rotation objective function, and $\mathcal{L}_{NTXent}$ is following Equation (1). More simply, instead of applying Equation (5) of the DDA, we apply Equation (6) for the SelfAugment to compare within our experiment. All other hyperparameters are kept the same.

We summarize the augmentation policy found by SelfAugment in Tables 5 and 6.

Table 3: List of all image augmentations that the policy can choose from during the search.

| Operation Name | Description | Range of magnitudes |
|---|---|---|
| Identical | No augmentation | N/A |
| Brightness | Adjust the brightness of the image. A magnitude of 0 does not modify the input image, whereas magnitude of 1 gives the white image. | [0.0, 1.0] |
| Contrast | Control the contrast of the image. A magnitude of 0 generates a completely black image, 1 does not modify the input image, while any other non-negative number modifies the brightness by this factor. | [0.0, 1.0] |
| Hue | The image hue is adjusted by converting the image to HSV and cyclically shifting the intensities in the hue channel (H). A magnitude of $\pi$ and $-\pi$ give complete reversal of hue channel in HSV space in positive and negative directions, respectively. 0 means no shift. | $[-\pi, \pi]$ |
| Saturation | Adjust the saturation of the image. A magnitude of 0 will give a black-and-white image, 1 will give the original image, and 2 will enhance the saturation by a factor of 2. | [0.0, 2.0] |
| Solarize | Invert all pixels above a threshold value of magnitude. | [0.0, 1.0] |
| Gaussian Blur | Blurs image with randomly chosen Gaussian blur. The kernel size is kept fixed at (23, 23), and the magnitude controls the standard deviation to be used for creating a kernel to perform blurring. | $[0.0, \infty]$ |
| Posterize | Reduce the number of bits for each pixel to magnitude bits. | [0, 8] |
| Gray | Convert the image to a grey scale. No magnitude parameters. | N/A |
| Sharpness | Adjust the sharpness of the image. Adjust the sharpness of the image. A magnitude of 0 gives the original image, whereas a magnitude of 1 gives the sharpened image. | [0.0, 1.0] |

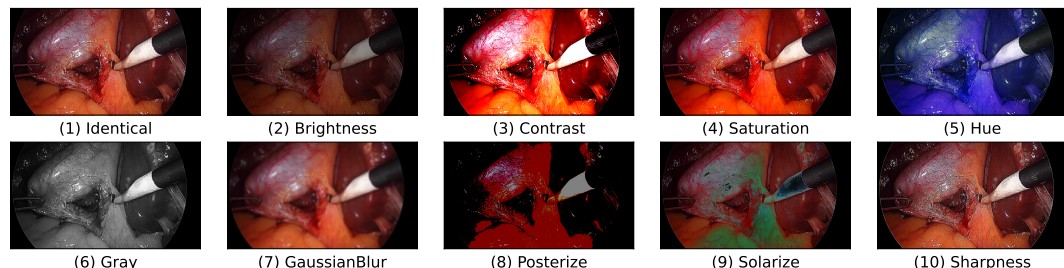

(1) Identical  (2) Brightness  (3) Contrast  (4) Saturation  (5) Hue
(6) Gray  (7) GaussianBlur  (8) Posterize  (9) Solarize  (10) Sharpness

Figure 5: Illustration of 10 augmentation operations (1)-(10) in our search space.

Table 4: The augmentation policy used by SimCLR

| | Augmentations | Strengths |
|---|---|---|
| Operation No.1 | Brightness (80%), Identical (20%) | 0.8 |
| Operation No.2 | Contrast (80%), Identical (20%) | 0.8 |
| Operation No.3 | Saturation (80%), Identical (20%) | 0.8 |
| Operation No.4 | Hue (80%), Identical (20%) | 1.26 |
| Operation No.5 | Gray (20%), Identical (80%) | N/A |
| Operation No.6 | GaussianBlur (50%), Identical (50%) | [0.0, 2.0] |

Table 5: The augmentation policy was found by SelfAugment(with $N = 5$) using our private SVHM dataset as the pretraining dataset.

| | Augmentations | Strengths |
|---|---|---|
| Operation No.1 | Hue (2%) | 1.11 |
| Operation No.2 | Contrast (97%), Hue (2%), Saturation (1%) | 0.01, 1.06, 1.98 |
| Operation No.3 | Brightness (95%), Hue (5%) | 0.32, 0.73 |
| Operation No.4 | Hue (100%) | 1.66 |
| Operation No.5 | Saturation (91%), Hue (7%), Brightness (1%), Contrast (1%) | 2.00, 1.11, 0.71, 0.25 |

Table 6: The augmentation policy was found by SelfAugment(with $N = 5$) using Cholec80 as the pretraining dataset.

|  | Augmentations | Strengths |
|---|---|---|
| Operation No.1 | Saturation(100%) | 2.00 |
| Operation No.2 | Saturation(100%) | 2.00 |
| Operation No.3 | Hue (100%) | 1.54 |
| Operation No.4 | Saturation (89%), Solarize (11%) | 2.00, 0.03 |
| Operation No.5 | Saturation (54%), Hue (25%), Solarize (14%) Contrast (7%) | 2.00, 2.57, 0.79, 0.09 |

### B.3. Augmentation Policy found by DDA

In this section, we summarize the augmentation policy found by our DDA. For our default choice $N = 5$, the found augmentation policy is summarized in Tables 7 and 8. The augmentation policies corresponding to Figure 2 are summarized in Figures 6 to 10.

Table 7: The augmentation policy was found by DDA(with $N = 5$) using our private SVHM dataset as the pretraining dataset.

|  | Augmentations | Strengths |
|---|---|---|
| Operation No.1 | Identical (89%), Posterize (8%), GaussianBlur (2%) | N/A, 0.96, [0.22, 0.28] |
| Operation No.2 | Saturation(66%), Sharpness (20%), Posterize (10%) | 1.07, 0.06, 0.99 |
| Operation No.3 | Identical (93%), Posterize (7%) | N/A, 1.00 |
| Operation No.4 | Identical (99%) | N/A |
| Operation No.5 | GaussianBlur (100%) | [0.17, 0.98] |

Table 8: The augmentation policy was found by DDA(with $N = 5$) using Cholec80 as the pretraining dataset.

|  | Augmentations | Strengths |
|---|---|---|
| Operation No.1 | Identical (54%), GaussianBlur (34%), Posterize (8%) | N/A, [0.16, 0.53], 1.00 |
| Operation No.2 | Saturation(90%), GaussianBlur (5%), Hue (4%) | 1.12, [0.14, 0.17], -1.32 |
| Operation No.3 | Identical (100%) | N/A |
| Operation No.4 | Identical (100%) | N/A |
| Operation No.5 | GaussianBlur (100%) | [0.17, 0.79] |

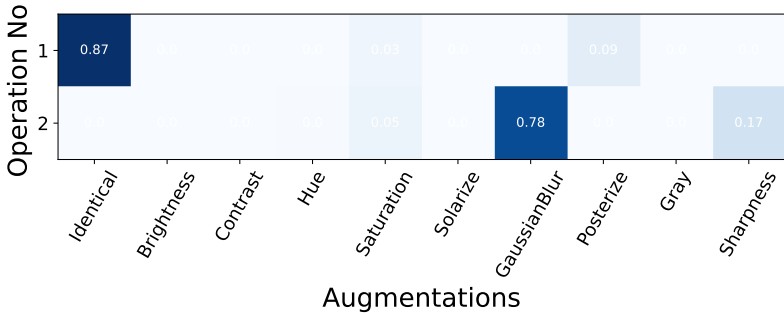

Figure 6: Augmentation policy found by DDA with $N = 2$ on the Cholec80 dataset. The operation number is indicated on y-axis, and augmentation choices on x-axis. The number on each cell indicates the probability of such augmentation being selected in corresponding operation.

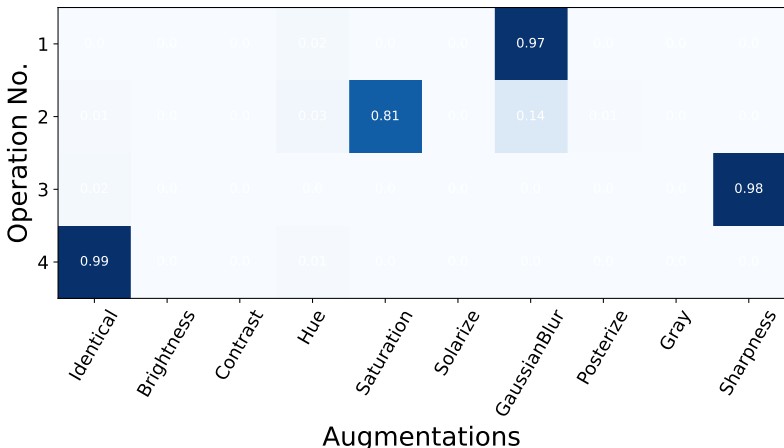

Figure 7: Augmentation policy found by DDAwith $N = 4$ on the Cholec80 dataset. The operation number is indicated on y-axis, and augmentation choices on x-axis. The number on each cell indicates the probability of such augmentation being selected in corresponding operation.

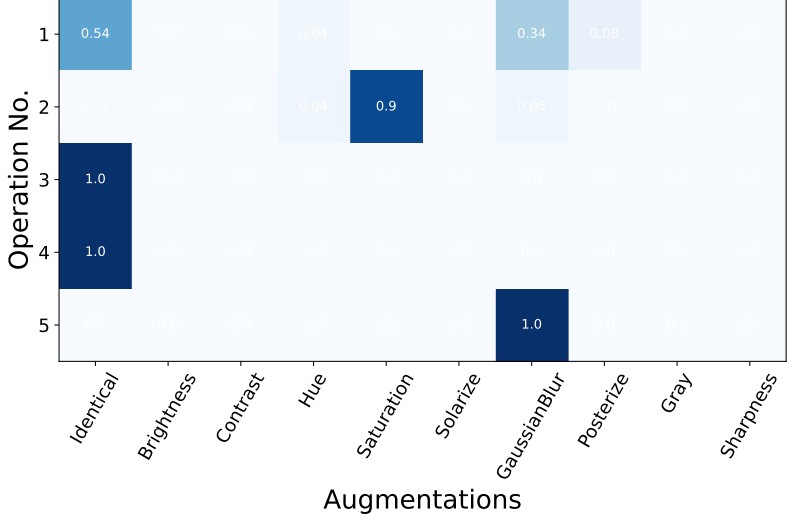

Figure 8: Augmentation policy found by DDA with $N = 5$ on the Cholec80 dataset. The operation number is indicated on y-axis, and augmentation choices on x-axis. The number on each cell indicates the probability of such augmentation being selected in corresponding operation.

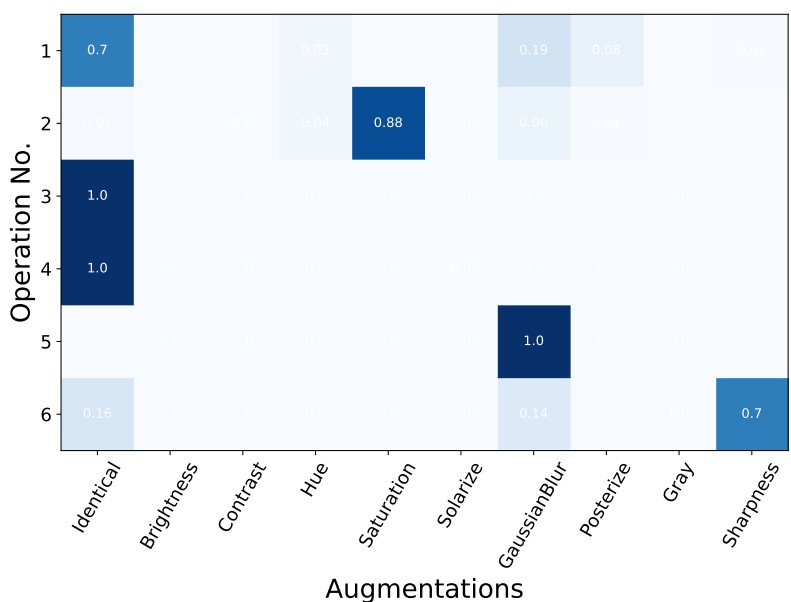

Figure 9: Augmentation policy found by DDA with $N = 6$ on the Cholec80 dataset. The operation number is indicated on y-axis, and augmentation choices on x-axis. The number on each cell indicates the probability of such augmentation being selected in corresponding operation.

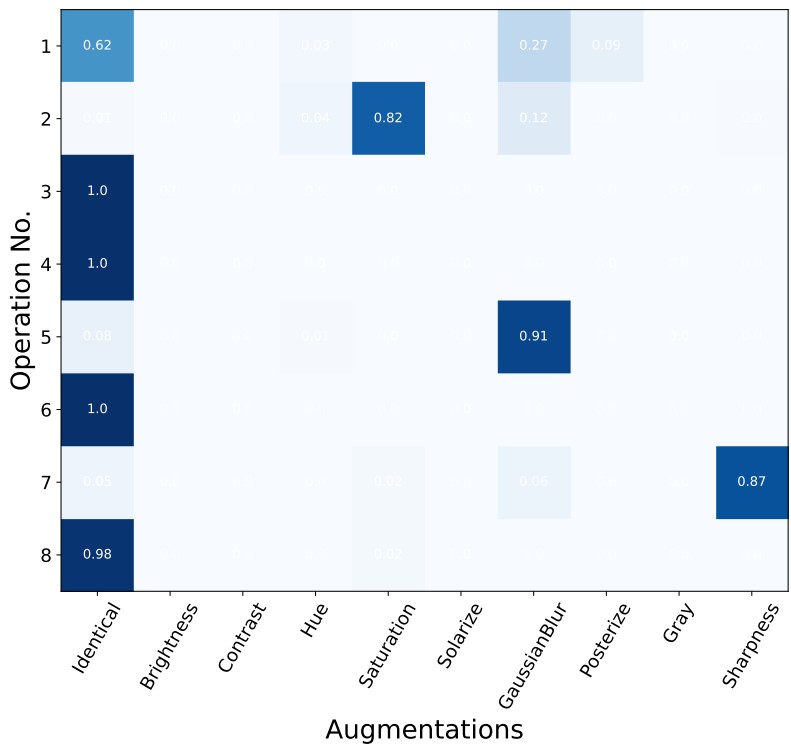

Figure 10: Augmentation policy found by DDA with $N = 8$ on the Cholec80 dataset. The operation number is indicated on y-axis, and augmentation choices on x-axis. The number on each cell indicates the probability of such augmentation being selected in corresponding operation.

### B.4. Additional Results

Table 9: Extended table of Table 1 with Argmax sampling. All results are based on using ResNet-50 as encoder and SimCLR as contrastive pretraining. Results of the linear probing are reported using mean Average Precision (mAP) (%), and finetuning on downstream segmentation tasks is reported using mIoU (%). The best results are in **boldface**.

| Pretraining Dataset | Augmentation | Sampling | Linear Prob (Classification) Cholec80 Tool | Finetune (Segmentation) SVHM Seg | CholecSeg8K |
|---|---|---|---|---|---|
| SVHM | SimCLR | N/A | 60.00 | 57.28 | 56.18 |
| | Base | N/A | 59.01 | 57.15 | **58.71** |
| | Random | Argmax | Complete Collapse | N/A | N/A |
| | Random | Categorical | Complete Collapse | N/A | N/A |
| | SelfAugment | Argmax | Complete Collapse | N/A | N/A |
| | SelfAugment | Categorical | Complete Collapse | N/A | N/A |
| | DDA | Argmax | 60.13 | 57.64 | 56.19 |
| | DDA | Categorical | **65.95** | **58.29** | 57.86 |
| Cholec80 | SimCLR | N/A | 67.59 | 58.29 | 56.02 |
| | Base | N/A | 67.78 | 58.36 | 57.04 |
| | Random | Argmax | Complete Collapse | N/A | N/A |
| | Random | Categorical | Complete Collapse | N/A | N/A |
| | SelfAugment | Argmax | 62.28 | 58.41 | 58.07 |
| | SelfAugment | Categorical | 60.24 | 58.41 | 55.86 |
| | DDA | Argmax | 72.02 | 58.86 | 55.09 |
| | DDA | Categorical | **73.59** | **59.31** | **59.40** |

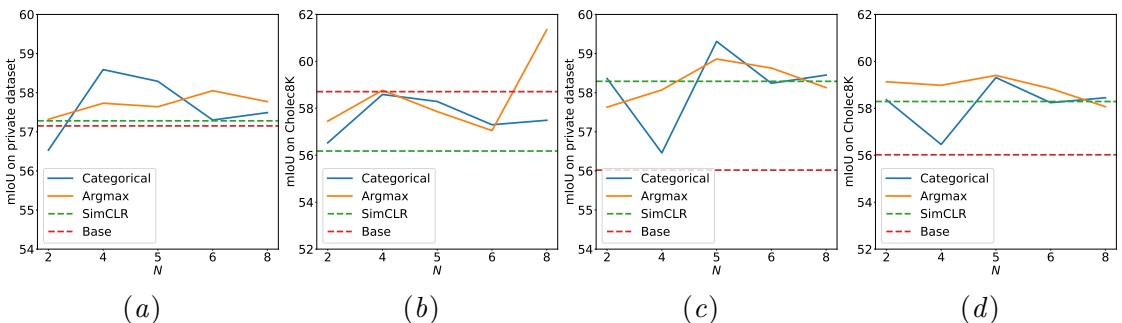

(a)  (b)  (c)  (d)

Figure 11: (a-b) Pretraining on our private dataset. Each data point is an individual run of the experiment, from augmentation search to pretraining and evaluations. (c-d) Pretraining on the public the dataset Cholec80. In subfigures (a) and (c), results are showing finetuning on our private SVHM Seg dataset. In subfigures (b) and (d), results are showing finetuning on the public dataset CholecSeg8k.

In Table 9, we present extended results for using the Argmax sampling for the final policy. It can be observed that sampling from categorical distributions results in better performance. As a result, we use sampling from categorical distributions as default for

Table 10: Extended table of Table 1 with MOCO contrastive pretraining. All results are based on using ResNet-50 as encoder. Results of the linear probing are reported using mean Average Precision (mAP) (%), and finetuning on downstream segmentation tasks is reported using mIoU (%). The best results are in **boldface**.

| Pretraining Dataset | Loss Objective | Augmentation | Linear Prob (Classification) | Finetune (Segmentation) | |
|---|---|---|---|---|---|
| | | | Cholec80 Tool | SVHM Seg | CholecSeg8K |
| SVHM | SimCLR | SimCLR | 60.00 | 57.28 | 56.18 |
| | SimCLR | DDA | **65.95** | **58.29** | **57.86** |
| | MoCo | MoCo | 53.94 | 57.03 | 58.60 |
| | MoCo | DDA | **57.35** | **57.81** | **58.91** |
| Cholec80 | SimCLR | SimCLR | 67.59 | 58.29 | 56.02 |
| | SimCLR | DDA | **73.59** | **59.31** | **59.40** |
| | MoCo | MoCo | 61.75 | 57.89 | 55.89 |
| | MoCo | DDA | **61.83** | **58.29** | **57.20** |

our DDA. Note that the SimCLR augmentation policy also uses sampling from categorical distributions (see Table 4).

In Figure 11, we plotted additional results for finetuning on CholecSeg8k and our Private Seg dataset. All details are the same as in Figure 2. It can be observed that it can either outperform or be on par with the SimCLR policy. We believe that the result in Figure 11(b) that slightly under-perform to the Base policy is due to the data distribution difference between our SVHM dataset and Cholec80. It is worth noting that slightly under-performing in a few finetuning tasks is common in SSL evaluations, and the main evaluation metric is the linear evaluations (Bardes et al., 2022; Huang et al., 2024). The goal of the SSL is learning general representations; on average, DDA demonstrates solid improvement over existing methods.

### B.5. Ablation Study on the Basic Augmentation for the Initial Encoder

For the initial encoder, we used image cropping due to its importance and effectiveness in contrastive learning. This has been studied in existing works on both natural images (Chen et al., 2020a) and X-ray images (Van der Sluijs et al., 2023). Other augmentations are also plausible. We performed an experiment with rotation as the initial augmentation choice. The experiment is conducted with the public dataset Cholec80, and the downstream evaluations are the same as the main paper. Results are in Table 11. It can be observed that using rotation can also outperform the baseline methods. However, image cropping as base for DDA is indeed more effective compared to rotation.

### B.6. Application on Other Datasets

Although in this paper, we mainly focused on the laparoscopic images, DDA can also be applied in other domains. In this subsection, we perform an experiment with an X-ray image dataset, CheXpert (Irvin et al., 2019) and the natural image dataset, CIFAR10 (Krizhevsky and Hinton, 2009). All experimental settings are the same as our main paper.

Table 11: All results are based on using ResNet-50 as encoder and SimCLR as contrastive pretraining. Results of the linear probing are reported using mean Average Precision (mAP) (%), and finetuning on downstream segmentation tasks is reported using mIoU (%). The best results are in **boldface**.

| Initial Augmentation | Augmentation | Linear Prob (Classification) | Finetune | |
| --- | --- | --- | --- | --- |
| | | Cholec80 Tool | SVHM Seg | CholecSeg8K |
| N/A | SimCLR | 67.59 | 58.29 | 56.02 |
| N/A | Base | 67.78 | 58.36 | 57.04 |
| Image Cropping | DDA | **73.59** | **59.31** | **59.40** |
| Rotation | DDA | 72.38 | 58.02 | 59.26 |

For CheXpert, we removed the operation converting to gray scale from the search space since X-ray images are already gray scale. Results reported using macro area under the ROC Curve (AUROC) with linear probing. For CIFAR10, the search space is the same as our main paper, and results are reported as classification accuacy. Results are in Table 12

Table 12: For CIFAR10, the results are reported with classification accuracy. For CheXpert, the results are reported as macro AUROC.

| Dataset | Augmentation | Linear Prob |
| --- | --- | --- |
| CheXpert | SimCLR | 72.4 |
| | DDA | 71.5 |
| CIFAR10 | SimCLR | 92.2 |
| | DDA | 91.6 |

It can be observed that DDA performs similarly with SimCLR on X-ray images. This does not indicate that DDA is not effective. This is because SimCLR policy is already performing in the optimal range. This dataset is also used by Van der Sluijs et al. (2023). They conducted a grid search for the optimal augmentation policy on X-ray images. By conducting grid-search, they tried different combinations to find the best one. They reported Macro AUROC with optimal augmentation operations on this dataset is in the range of 68.8 to 73.6. This indicates that the SimCLR augmentation policy behaves differently with X-ray images compared to laparoscopic images. This result also indicates that DDA can find suitable augmentation for X-ray images. In the SimCLR paper (Chen et al., 2020a), the authors also conducted a grid search on CIFAR10 to find the optimal policy. It can be observed that DDA can also find suitable augmentation for natural images.

In existing works (Chen et al., 2020a; Van der Sluijs et al., 2023) that conduct a grid search, one needs to perform the pretraining on every possible combination of the augmentation. For DDA, it just needs the pretraining once, regardless of the number of combinations. Although the DDA result might not be the best one, it is very close to the best-performing one. The efficiency of DDA makes it very suitable for other medical images, which often have different characteristics.

## Appendix C. Visualisations of the Augmented Images

In this section, we show the augmented images. For a batch of randomly selected images with no augmentation applied in Figure 12, Figure 13 shows the augmented images using our DDA, Figure 14 shows the augmented images using SimCLR policy, Figure 15 shows the augmented images using SelfAugment augmentation, and Figure 16 shows the augmented images using random augmentation. To demonstrate the overlapping view we discussed in Section 3.2, for every image shown in Figures 13 and 14, considering how easily to find another image that is visually similar to itself. Since laparoscopic cholecystectomy (LC) dataset contents are predominantly red in colour, a green gallbladder in the first row, the second and third columns from the top left corner of Figure 14, is very unlikely to match with other images. As a result, augmentation operations like *Hue* that change colour profiles are unsuitable in LC. Augmentation policies found by DDA(summarized in Section B.3) are inherently more suitable for the dataset it is searched on.

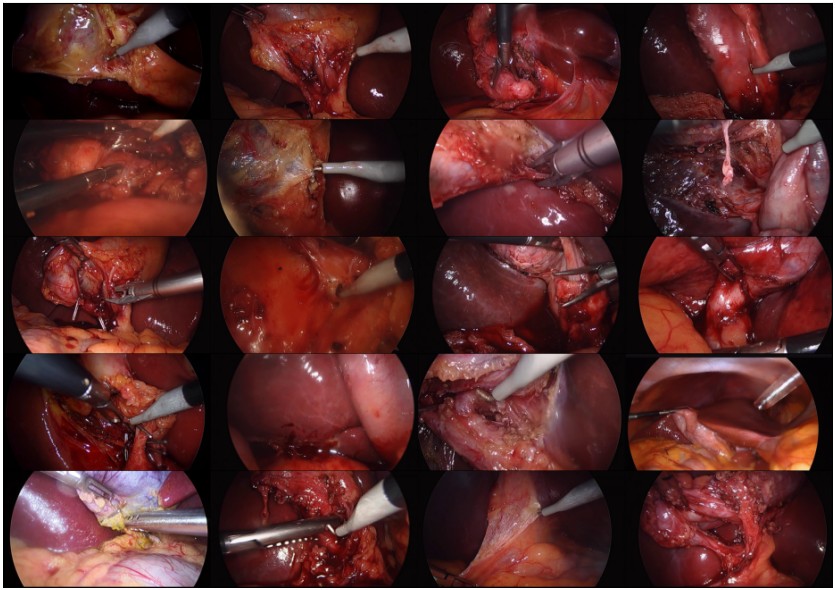

Figure 12: Visualisations of the images in our private dataset with no augmentation applied.

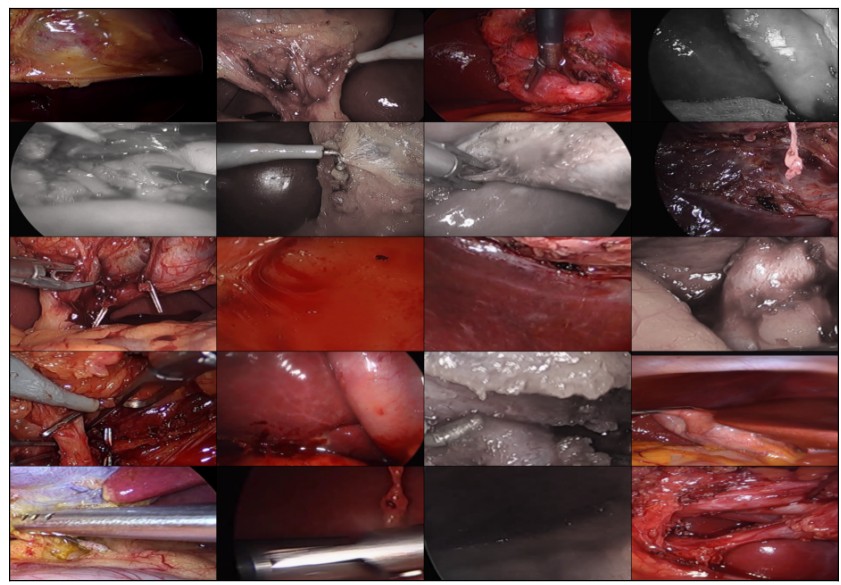

Figure 13: Visualisations of the images in our private dataset with the augmentation policy found by DDA($N = 5$).

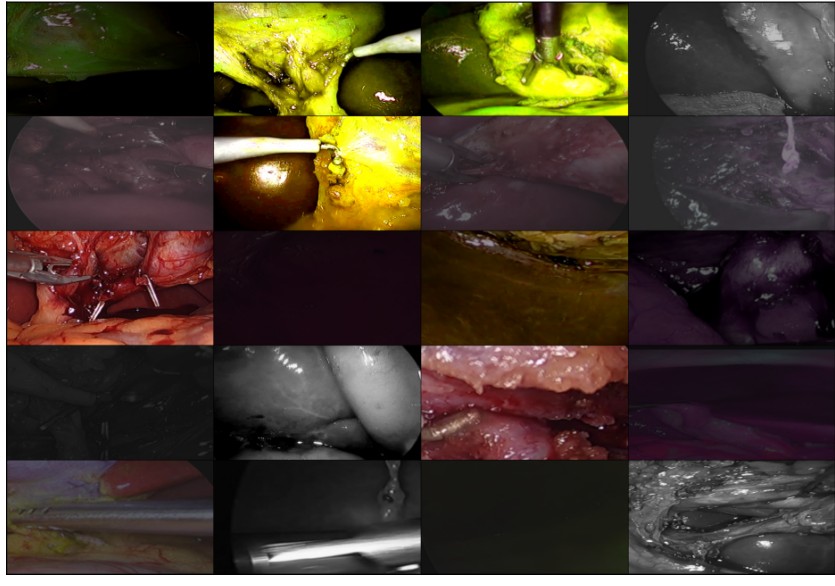

Figure 14: Visualisations of the images in our private dataset with the SimCLR augmentation policy.

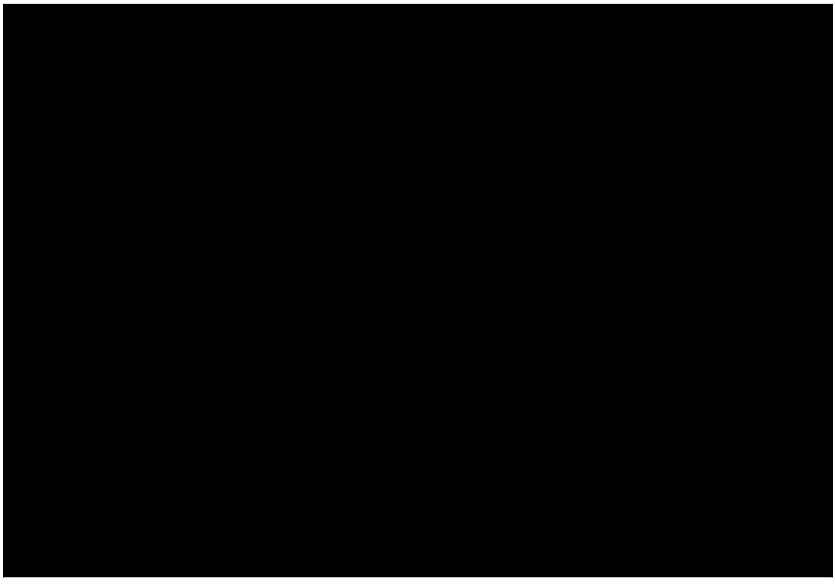

Figure 15: Visualisations of the images in our private dataset with the SelfAugment augmentation applied.

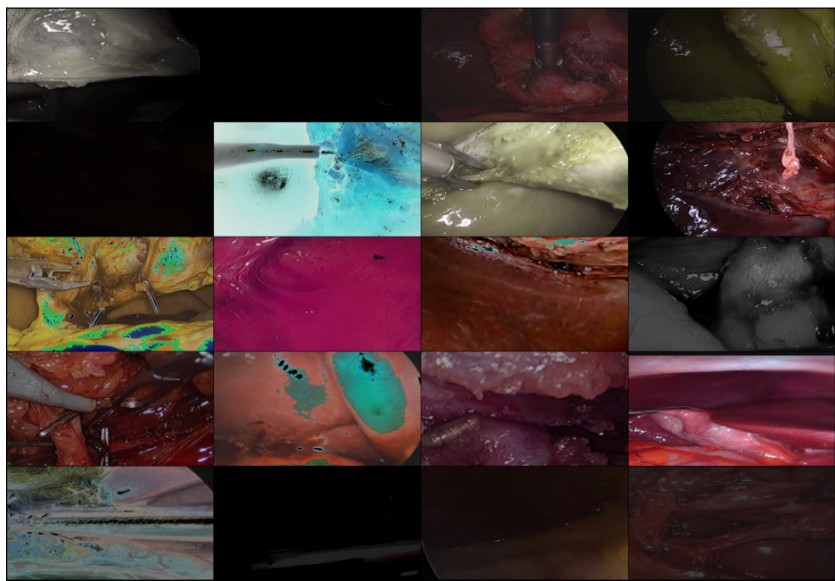

Figure 16: Visualisations of the images in our private dataset with random augmentation applied.

