# OpenReview forum: "DDA: Dimensionality Driven Augmentation Search for Contrastive Learning in Laparoscopic Surgery"
_MIDL.io/2024/Conference — MIDL 2024 Poster_

### Official Review · Reviewer_wQKj · 2024-02-27

**Confidence:** 4
**Preliminary Rating:** 4
**Recommendation:** Oral

**Summary:**

An important consideration in SSL is the choice of data augmentation, which often depends on the specific domain. It is unclear whether commonly used augmentation policies for general image types are suitable for surgical applications. Authors propose to automate the search for a high performing combination of augmentations to achieve good contrastive learning results. They name their method as Dimensionality Driven Augmentation Search (DDA). DDA utilizes the local dimensionality of deep representations as a proxy target and explores appropriate data augmentation strategies within the framework of contrastive learning.

**Strengths:**

The drawback of the the existing methods in the literature is well framed and need for the proposed model is well presented.

DDA reduces  the need for annotated datasets to determine the optimal set of augmentations for SSL

The authors provide a comprehensive set of experimental results.

**Weaknesses:**

Supervised baseline performance if not presented.

Looking at the Figures 5-9 in appendix, it seems like DDA can pick the same augmentation multiple times. This is an advantage of the DDA, but also makes the comparisons against the other methods a bit unfair as they are forced to use harming augmentations.

Behavior of DDA illustrated in Fig 8-9 in appendix B is confusing for some experiments

**Detailed Comments:**

Even if DDA outperforms all the other methods that it is compared against, it is not clear if the final model is working at a meaningful performance range. Comparison against a  supervised model is required to understand where the SSL models stay w.r.t this performance bound

The necessity of initially acquiring an encoder trained with "most basic augmentation" augmentation, such as image cropping, before implementingDDA, lacks clarity. One may question whether the LID-based model can function effectively without these preliminary step.

Is it possible for the authors to also show statistics on which augmentations are picked by the SelfAugment.

Looking at the Figures 5-9 in appendix, it seems like DDA can pick the same augmentation multiple times. This is an advantage of the DDA over random, but also makes the comparisons against the other methods a bit unfair as they are forced to use harming augmentations.  Can the authors conduct an experiment with only using a subset augmentations that they found to be most beneficial. This may beat the purpose of the study a bit as the selection is done manually, but can be a good experiment to understand of the cause of the "Collapses".

It is also not clear why the "Random" augmentation model always collapsed. Can the reason be a limited search space?

Behavior of DDA illustrated in Fig 8-9 in appendix B is confusing. Can the authors elaborate on, why "sharpness" augmentation picked to be used by DDA after 2 "identical" augmentation steps.

**Justification Of The Preliminary Rating:**

The paper is well written and the proposed methodology has good theoretical support.

Extensive experiments are presented to help the readers understand and evaluate the presented method.

Some comparisons and results require further explanation to evaluate the performance

**Questions To Address In The Rebuttal:**

Please present the supervised baseline model performance to see where the self-supervised model stands wrt that baseline.

Is it possible for the authors to also show statistics on which augmentations are picked by the SelfAugment.

Can DDA work effectively without these initial training on basic augmentations.

Can the authors conduct an experiment with only using a subset augmentations that they found to be most beneficial. This may beat the purpose of the study a bit as the selection is done manually, but can be a good experiment to understand of the cause of the "Collapses".

**Special Issue:**

Yes

---

> ### Author Response · Authors · 2024-03-18
> **Response to Reviewer wQKj part 1**
>
> Thanks for your insightful reviews. Please find our response to your questions below:
>
> ---
>
> **Q1:** Supervised baseline performance
>
> **A1:** We have revised the draft to incorporate the results of supervised learning (i.e., without self-supervised learning pretraining). In the Cholec80 Tool classification results, for the supervised baseline, we employed end-to-end supervised training instead of linear probing. It's worth noting that linear probing is the standard evaluation protocol typically reserved for self-supervised learning [1, 2, 3]. Please find the results below:
>
> | Pretraining Dataset | Augmentation | Linear Prob (Classification) | Finetune | Finetune |
> |:-------------------:|:------------:|:-------------------------------:|:-----------:|:-----------:|
> |                     |              |          Cholec80 Tool          |   SVHM Seg  | CholecSeg8K |
> |  None (Supervised)  |  Supervised  |              49.95              |    55.77    |    57.79    |
> |         SVHM        |    SimCLR    |              60.00              |    57.28    |    56.18    |
> |         SVHM        |     Base     |              59.01              |    57.15    |  **58.71**  |
> |         SVHM        |    Random    |             Collapse            |     N/A     |     N/A     |
> |         SVHM        |  SelfAugment |             Collapse            |     N/A     |     N/A     |
> |         SVHM        |      DDA     |            **65.95**            |  **58.29**  |    57.86    |
> |       Cholec80      |    SimCLR    |              67.59              |    58.29    |    56.02    |
> |       Cholec80      |     Base     |              67.78              |    58.36    |    57.04    |
> |       Cholec80      |    Random    |             Collapse            |     N/A     |     N/A     |
> |       Cholec80      |  SelfAugment |              60.24              |    58.41    |    57.04    |
> |       Cholec80      |      DDA     |            **73.59**            |  **59.31**  |  **59.40**  |
>
> ---
>
> **Q2:**  Basic augmentation lacks clarification
>
> **A2:** The initial encoder plays a crucial role in extracting representations utilized for the search process. This component is also integral to the SelfAugment method, which necessitates an additional layer atop the initial encoder. Notably, DDA does not require any supplementary layer.
>
> Image cropping is selected for its significance and efficacy in contrastive learning, which has been extensively explored in studies on both natural images [1] and X-ray images [4]. Other augmentations are also plausible. We performed an experiment with rotation as the initial augmentation choice. The experiment was conducted with the public dataset Cholec80, with downstream evaluations consistent with those outlined in the main paper. It can be observed that rotation also performs similarly to image cropping as the basic augmentation for the initial encoder, and DDA can outperform the baseline regardless of the initial choice of the augmentation.
>
> Results are in table below:
>
> | Initial Augmentation | Augmentation | Linear Prob (Classification) | Finetune | Finetune |
> |:--------------------:|:------------:|:-------------------------------:|:-----------:|:-----------:|
> |                      |              |          Cholec80 Tool          |   SVHM Seg  | CholecSeg8K |
> |          N/A         |    SimCLR    |              67.59              |    58.29    |    56.02    |
> |          N/A         |     Base     |              67.78              |    58.36    |    57.04    |
> |    Image Cropping    |      DDA     |            **73.59**            |  **59.31**  |  **59.40**  |
> |       Rotation       |      DDA     |              72.38              |    58.02    |    59.26    |

---

> ### Author Response · Authors · 2024-03-18
> **Response to Reviewer wQKj part 2**
>
> **Q3:** Show statistics on which augmentations are picked by the SelfAugment
>
> **A3:**  The augmentation policy found by SelfAugment (with N = 5) using our private SVHM dataset as the pretraining dataset is shown below.
>
> |                |                          Augmentations                         |        Strengths       |
> |:--------------:|:--------------------------------------------------------------:|:----------------------:|
> | Operation No.1 |                            Hue (2%)                           |          1.11          |
> | Operation No.2 |          Contrast (97%), Hue (2%), Saturation (1%)          |    0.01, 1.06, 1.98    |
> | Operation No.3 |                  Brightness (95%), Hue (5%)                  |       0.32, 0.73       |
> | Operation No.4 |                           Hue (100%)                          |          1.66          |
> | Operation No.5 | Saturation (91%), Hue (7%), Brightness (\%), Contrast (1%) | 2.00, 1.11, 0.71, 0.25 |
>
> The augmentation policy found by SelfAugment (with N = 5) using Cholec80 as the pretraining dataset is shown below.
>
> |                |                         Augmentations                         |        Strengths       |
> |:--------------:|:-------------------------------------------------------------:|:----------------------:|
> | Operation No.1 |                       Saturation(100%)                       |          2.00          |
> | Operation No.2 |                       Saturation(100%)                       |          2.00          |
> | Operation No.3 |                          Hue (100%)                          |          1.54          |
> | Operation No.4 |               Saturation (89%), Solarize (11%)              |       2.00, 0.03       |
> | Operation No.5 | Saturation (54%), Hue (25%), Solarize (14%) Contrast (%) | 2.00, 2.57, 0.79, 0.09 |
>
> We have added these statistics in the paper Appendix B.2. Tables 5 and 6.
>
> ---
>
> **Q4:** DDA can pick the same augmentation multiple times, which could be unfair to other methods
>
> **A4:** This is not the case. In our comparison with SelfAugment, we utilized the exact same search space (initial pool of augmentations for selection). However, the outcome of SelfAugment's selected policy resulted in dimensional collapse. This observation also extends to randomly selected augmentations (Random Policy), which are chosen from the same search space.
>
> Both SelfAugment and Random policies have the ability to select identical operations multiple times. However, SelfAug neither selects beneficial augmentations nor identical ones. For the random policy, the repetition of identical operations is naturally a rare occurrence.
>
> For SimCLR, the augmentation is the one selected by the authors of the original paper, who conducted a grid search using natural images. They determined their optimal augmentation policy through supervised evaluation with labeled data. As discussed in Section 2.2, conducting a two-stage grid search can be exceedingly time-consuming and requires additional labeled data. To provide context, within our search space comprising 10 augmentation operations, identifying a policy with 5 operations on SVHM would demand $10^5 \times 42 = 4,200,000 $ hours (42 hours for contrastive pretraining and finetuning) using grid search. Given the constraints of time and computational resources, it is impractical for us to replicate the grid search on our datasets.
>
> In summary, we did not force any baseline to select harmful augmentations.
>
> ---

---

> ### Author Response · Authors · 2024-03-18
> **Response to Reviewer wQKj part 3**
>
> **Q5:** Experiments with the most beneficial subset augmentations
>
> **A5:** Based on our experimental findings with DDA, we have conducted an additional experiment where we exclusively utilize augmentation operations that DDA found to be suitable for laparoscopic images. These augmentations constitute the initial pool for the search and include Identical, Brightness, Contrast, Saturation, Gaussian Blur, Posterize, and Sharpness. We removed the Hue, Solarize, and Gray Scale since they alter color profiles. It's worth noting that the selection of these operations would not have been possible without the utilization of DDA. We refer to the original search space as **S1** and this refined search space as **S2**.
>
> The results are in the following Table.
>
> | Pretraining Dataset | Search Space | Augmentation | Linear Prob (Classification) |  Finetune |   Finetune  |
> |:-------------------:|:------------:|:------------:|:----------------------------:|:---------:|:-----------:|
> |                     |              |              |         Cholec80 Tool        |  SVHM Seg | CholecSeg8K |
> |         SVHM        |       -      |     Base     |             60.00            |   57.28   |    56.18    |
> |         SVHM        |       -      |    SimCLR    |             59.01            |   57.15   |  **58.71**  |
> |         SVHM        |      S1      |    Random    |           Collapse           |    N/A    |     N/A     |
> |         SVHM        |      S1      |  SelfAugment |           Collapse           |    N/A    |     N/A     |
> |         SVHM        |      S1      |      DDA     |           **65.95**          | **58.29** |    57.86    |
> |         SVHM        |      S2      |    Random    |           Collapse           |    N/A    |     N/A     |
> |         SVHM        |      S2      |  SelfAugment |           Collapse           |    N/A    |     N/A     |
> |         SVHM        |      S2      |      DDA     |             62.23            |   57.57   |    58.51    |
> |         ---         |      ---     |      ---     |              ---             |    ---    |     ---     |
> |       Cholec80      |       -      |     Base     |             67.59            |   58.29   |    56.02    |
> |       Cholec80      |       -      |    SimCLR    |             67.78            |   58.36   |    57.04    |
> |       Cholec80      |      S1      |    Random    |           Collapse           |    N/A    |     N/A     |
> |       Cholec80      |      S1      |  SelfAugment |             60.24            |   58.41   |    55.86    |
> |       Cholec80      |      S1      |      DDA     |           **73.59**          | **59.31** |    59.40    |
> |       Cholec80      |      S2      |    Random    |           Collapse           |    N/A    |     N/A     |
> |       Cholec80      |      S2      |  SelfAugment |           Collapse           |    N/A    |     N/A     |
> |       Cholec80      |      S2      |      DDA     |             72.08            |   58.13   |  **60.51**  |
>
>
> It can be observed that changing the search space does not affect the performance of DDA. DDA consistently demonstrates significant improvements in linear probing, which serves as the primary evaluation metric for existing SSL works. Furthermore, we investigated the policies selected within search space S2. In S2, DDA demonstrates a preference for selecting Identical, Gaussian Blur, and Saturation augmentations. This choice remains consistent with the selection observed in search space S1, albeit with slight variations in the strength and sequence of operations. Conversely, both Random policies and SelfAugment once again result in complete collapse.

---

> ### Author Response · Authors · 2024-03-18
> **Response to Reviewer wQKj part 4**
>
> **A5** (continued):
>
> Below tables show the policies found by DDA and SelfAugment in S2.
>
> The augmentation policy found by DDA (with N = 5) using Cholec80 as the pretraining dataset is shown below.
>
> |                | Augmentations                                                          | Strengths                    |
> |----------------|------------------------------------------------------------------------|------------------------------|
> | Operation No.1 | GaussianBlur (48%), Identical (23%), Saturation (21%), Posterize (8%) | [0.15,0.16], N/A, 1.23, 1.00 |
> | Operation No.2 | Identical (100%)                                                       | N/A                          |
> | Operation No.3 | Identical (100%)                                                       | N/A                          |
> | Operation No.4 | Identical (100%)                                                       | N/A                          |
> | Operation No.5 | GaussianBlur (84%), Sharpness (16%)                                    | [0.84, 1.45], 0.97        |
>
> The augmentation policy found by SelfAugment (with N = 5) using Cholec80 as the pretraining dataset is shown below.
>
> |                | Augmentations                                                       | Strengths                      |
> |----------------|---------------------------------------------------------------------|--------------------------------|
> | Operation No.1 | Sharpness (96%), Saturation (2%), GaussianBlur (1%), Posterize (1%) | 0.75, 1.25, [0.23, 0.59], 0.68 |
> | Operation No.2 | Posterize (68%), Brightness (29%), GaussianBlur (3%)                | 0.78, 0.21, [0.63, 1.01]       |
> | Operation No.3 | Contrast (98%), Sharpness (2%)                                      | 0.42, 0.70                     |
> | Operation No.4 | Brightness (82%), Contrast (17%), GaussianBlur (1%)                 | 0.39, 0.52, [0.11, 0.12]       |
> | Operation No.5 | Posterize (100%)                                                    | 1.00                           |
>
> ---
>
> **Q6:** Why the "Random" augmentation model always collapsed. Can the reason be a limited search space?
>
> **A6:** We speculate that this phenomenon arises from the variance induced by augmentation being greater than the variance stemming from the data distribution itself. A similar observation has been noted in a previous study [5]. In laparoscopic surgery, images typically contain anatomical tissue and organs, where red hues are prevalent while other colors are considerably less common. This suggests a limited variance in the data distribution within the pixel space. Excessive augmentation may lead to an increased variance, potentially triggering collapse.
>
> To further illustrate this concept, we have included visualizations of randomly selected batches of images from the SVHM dataset with various augmentation policies applied. These visualizations can be found in Appendix C, Figures 11-15, for reference.
>
> ---
>
> **Q7:** Behavior of DDA illustrated in Fig 8-9 in appendix B is confusing for some experiments. Why "sharpness" augmentation is picked to be used by DDA after 2 "identical" augmentation steps?
>
> **A7:** It is indeed what DDA chose. An explanation could be the optimizations and initializations of parameters. Different parameter initializations may lead to varying operations in different sequences. However, we have found that DDA consistently tends to select the same set of operations, irrespective of parameter initializations or the number of operations to be selected.
>
> Figures 5-9 were intended to depict the exact augmentation choice found by DDA for varying numbers of augmentation operations ($N$). The statistics of each particular operation are provided in Figure 3 (c-d). These Figures (5-9 and 3) are intended to show which operations are more likely to be selected by DDA; the exact augmentation choices may vary based on the parameter initializations. Statistically, DDA demonstrates a preference for the augmentations depicted in Figure 3 (c-d).
>
> In the revised version, we have updated the captions for Figures 5-9. The x-axis represents the operation, the y-axis represents the index in the sequence of the policy, and the number in each cell indicates the probability, with darker shades (blue) indicating higher probabilities.
>
> --------------------------------------------------------------------
>
> [1] Chen, Ting, et al. "A simple framework for contrastive learning of visual representations." ICML 2020.\
> [2] Grill, Jean-Bastien, et al. "Bootstrap your own latent-a new approach to self-supervised learning." NeurIPS 2020.\
> [3] Bardes, Adrien, Jean Ponce, and Yann LeCun. "VICReg: Variance-Invariance-Covariance Regularization for Self-Supervised Learning." ICLR 2021.\
> [4] Van der Sluijs, Rogier, et al. "Exploring image augmentations for siamese representation learning with chest x-rays." MIDL, 2023.\
> [5] Jing, Li, et al. "Understanding Dimensional Collapse in Contrastive Self-supervised Learning." ICLR 2022.

---

> ### Author Response · Authors · 2024-03-25
> **Thanks for your feedback**
>
> Dear reviewer wQKj,
>
> Thank you so much for recognising our paper and providing us with such a great opportunity to improve it further. Please kindly let us know if we have addressed your questions, and we would be grateful for any further feedback.

---

### Official Review · Reviewer_U9nn · 2024-03-05

**Confidence:** 5
**Preliminary Rating:** 3
**Final Rating:** 4

**Summary:**

The paper proposes optimizing an augmentation policy from among a set of manually selected possible augmentations, to improve the representation of contrastive self-supervised pretraining. The authors study their proposed method, Dimensionality Driven Augmentation (DDA) with SimCLR method in two surgical datasets, and show that the method mostly improves downstream classification and segmentation compared to randomly sampled augmentations and original SimCLR augmentation.

**Strengths:**

- The paper is mostly written well and easy to read.
- Finding the right set of augmentations during self-supervised pretraining is challenging, but a significant problem that authors try to tackle, proposing a novel method.
- Optimizing augmentation policies during retraining by optimizing in the feature space of the internal representation without requiring labels is attractive.

**Weaknesses:**

- The paper’s experiments with only SimCLR and only photographic medical image is limiting assessment of how the method’s performance (and assumptions) hold for other popular contrastive learning methods and for radiology images such as X-ray or ultrasound.
- The baseline’s performance depends on the author’s choice of set of possible augmentation. A better baseline would be a set of reasonable augmentations known in the literature for the target dataset (or types of images) coming from domain knowledge (e.g. choosing from semantics-preserving set of augmentations)? Or previous work where optimal performance has been achieved in classification and segmentation through manual selection of sensible augmentations during pretraining of the target dataset.

**Detailed Comments:**

Sec 2.3 could be improved for readability and explaining the concept of the query data, feature space, number of data and distance in the context of the surgical images.
It starts a bit difficult to understand from “To measure if query data is well represented …, growth rate …. As distance increases”. Then to describe LID degrees of freedom and the local neighborhood of query points are used. But most of these are not defined and the abstraction is difficult to put into context. A better way could be to start from the example of the surgical images and describe what query points are, notion of neighborhood and distance and degrees of freedom, then explain growth rate and LID within that context. Then finally provide the generalized notion when setting up for the Theorem 1.

Ablation on initial condition? i.e. starting with augmentation different than cropping? How much this impacts the performance?

Minor points and typos:
- Sec 1, contributions, “surgical LC datasets” what’s LC ?
- Table 1 caption: “pertaining”
- Figure 4: “simmilar”

**Justification Of Final Rating:**

The authors have adequately addressed the comments; the paper is valuable and interesting for the community working in endoscopic datasets, and those working in developing better ways to automatically identify best augmentation policies in contrastive SSL training. I've upgraded the rating.

**Justification Of The Preliminary Rating:**

The paper proposes an interesting method for an important problem, is written well mostly, and provides results that might be promising but lacks a more suitable baseline, more comprehensive experiments on different contrastive methods and types of datasets.

**Questions To Address In The Rebuttal:**

Comments in the Weaknesses.

---

> ### Author Response · Authors · 2024-03-18
> **Response to Reviewer U9nn part 1**
>
> Thank you very much for your valuable comments. We hope the following clarifications can help address the raised issues.
>
> ---
>
> **Q1:** Other contrastive learning methods
>
> **A1:** We have added an experiment including an additional popular contrastive learning framework, MoCo [1]. We compare the augmentation policy found by our DDA with the augmentation policy used by MoCo. Please see the results in the following table. We also included the results from our main paper as a reference.
>
> We use the policy found by DDA in the main paper for MoCo evaluations. It can be observed that the policy transfers well to MoCo on the SVHM dataset by substantially boosting classification accuracy and giving small increases in finetuning performance. On Cholec80, it is marginally less effective for the linear probing. We believe this might be due to the initial encoder used for the search being trained with SimCLR. In the future, we will add another experiment that uses MoCo with basic augmentation for the initial encoder.
>
> | Pretraining Dataset | Loss Objective | Augmentation | Linear Prob (Classification) |  Finetune |   Finetune  |
> |:-------------------:|:--------------:|:------------:|:----------------------------:|:---------:|:-----------:|
> |                     |                |              |         Cholec80 Tool        |  SVHM Seg | CholecSeg8K |
> |         SVHM        |     SimCLR     |    SimCLR    |             60.00            |   57.28   |    56.18    |
> |         SVHM        |     SimCLR     |      DDA     |           **65.95**          | **58.29** |  **57.86**  |
> |         SVHM        |      MoCo      |     MoCo     |             53.94            |   57.03   |    58.60    |
> |         SVHM        |      MoCo      |      DDA     |           **57.35**          | **57.81** |  **58.91**  |
> |       Cholec80      |     SimCLR     |    SimCLR    |             67.59            |   58.29   |    56.02    |
> |       Cholec80      |     SimCLR     |      DDA     |           **73.59**          | **59.31** |  **59.40**  |
> |       Cholec80      |      MoCo      |     MoCo     |             61.75            |   57.89   |    55.89    |
> |       Cholec80      |      MoCo      |      DDA     |           **61.83**          | **58.29** |  **57.20**  |

---

> ### Author Response · Authors · 2024-03-18
> **Response to Reviewer U9nn part 2**
>
> **Q2:** Other medical dataset (radiology images such as X-ray or ultrasound)
>
> **A2:**
> Although X-ray data is outside the scope of our study (our focus is on RGB data in the context of laparoscopic surgery), we conducted an additional experiment using an X-ray dataset, the CheXpert [2]. All experimental settings remained consistent with those outlined in our main paper, with the exception of removing the operation converting RGB inputs to grayscale versions from the search space, as X-ray images are inherently grayscale. Please find the results in the table below, which are reported using Macro AUROC (area under the ROC Curve) with linear probing.
>
> | Augmentation | Linear Prob |
> |:------------:|:-------------------------------:|
> |    SimCLR    |               72.4              |
> |      DDA     |               71.5              |
>
> It can be observed that DDA performs similarly (but with a little lower accuracy) to SimCLR on X-ray images. However, this similarity does not imply ineffectiveness of DDA. Rather, it likely suggests that the SimCLR policy is already operating within an optimal range for this type of dataset. Van der Sluijs, Rogier, et al. [3], who also utilized this dataset, conducted a grid search to determine the optimal augmentation policy for X-ray images. Through exhaustive exploration, they identified augmentation policies yielding Macro AUROC scores ranging from 68.8 to 73.6. This variance indicates that the SimCLR augmentation policy behaves differently with X-ray images compared to laparoscopic images. Moreover, these findings suggest that DDA is capable of discovering suitable augmentations for X-ray images.
>
>
> Additionally, we extended our experimentation to the CIFAR10 dataset, commonly employed for natural images. In the SimCLR paper [4], the authors conducted a grid search to find the optimal policy for natural images. Please find our results below, results are reported with linear probing classification accuracy.
>
> | Augmentation | Linear Prob |
> |:------------:|:-------------------------------:|
> |    SimCLR    |               92.2              |
> |      DDA     |               91.6              |
>
> Again, it can be observed that DDA can also find suitable augmentation (but with a little lower accuracy) for natural images.    Note that SimCLR requires supervised validation data to be used in grid search, whilst DDA does not.
>
> Please note that in conducting a grid search, one needs to perform the pretraining on every possible combination of the augmentation and subsequent supervised finetuning for evaluation. In contrast, DDA needs pretraining only once, regardless of the number of combinations, and doesn’t require any labeled datasets for selection. Although the DDA result might not always achieve the best performance, it consistently approaches the optimal outcome closely. The efficiency of DDA makes it highly suitable for various medical images, which often exhibit diverse characteristics.

---

> ### Author Response · Authors · 2024-03-18
> **Response to Reviewer U9nn part 3**
>
> **Q3:** Better baseline of augmentation policies (with domain knowledge)?
>
> **A3:** To our knowledge, there are currently no existing augmentation studies specifically tailored for self-supervised learning of laparoscopic images. Regarding supervised learning, previous works [5, 6, 7, 8] have demonstrated that rotation by 30 degrees, contrast adjustment, Gaussian noise, and Gaussian blurring are commonly used for segmentation tasks. Leveraging this domain knowledge, we constructed an additional manual selection policy using these popular augmentations from the literature. Each augmentation has a probability of 0.8 for application. Specifically, rotation strength is set at 30 degrees, while contrast and Gaussian noise strengths are randomly sampled. For Gaussian blur, sigma values range from 0.1 to 2.0, as per settings outlined in the aforementioned literature.
>
> Please refer to the results below, which also included outcomes from the main paper for reference. From the findings, it is evident that incorporating manually selected augmentations based on domain knowledge does not establish a strong baseline for contrastive learning.
>
> | Pretraining Dataset | Augmentation | Linear Prob (Classification) | Finetune |  Finetune |
> |:-------------------:|:------------:|:-------------------------------:|:----------:|:-----------:|
> |                     |              |          Cholec80 Tool          |  SVHM Seg  | CholecSeg8K |
> |         SVHM        |    SimCLR    |              60.00              |    57.28   |    56.18    |
> |         SVHM        |     Base     |              59.01              |    57.15   |    **58.71**    |
> |         SVHM        |    Manual    |              47.97              |    54.09   |  57.11  |
> |         SVHM        |      DDA     |            **65.95**            |  **58.29** |    57.86    |
> |       Cholec80      |    SimCLR    |              67.59              |    58.29   |    56.02    |
> |       Cholec80      |     Base     |              67.78              |    58.36   |    57.04    |
> |       Cholec80      |    Manual    |              59.00              |    55.12   |    59.01    |
> |       Cholec80      |      DDA     |            **73.59**            |  **59.31** |  **59.40**  |
>
>
> ---
>
> **Q4**: Sec 2.3 explain the terms in the context of surgical data
>
> **A4**: We have edited Section 2.3 accordingly, and provided a visualization of the local neighbourhood of a query data projected into a 2 dimensional space (Appendix A, Figure 4.). In the context of contrastive learning, one might view LID as the minimal number of features (dimensions) required to encapsulate the neighborhood of a single medical image in comparison to other images within the representation space. Given that laparoscopic surgery images frequently feature similar organs, instruments, and viewpoints, distinguishing between them becomes challenging. In contrastive learning, negative pairs (augmented views from different images) should exhibit significant dissimilarity. To effectively discriminate between representations of different laparoscopic images, it is intuitive to expect that a greater number of features (dimensions) are necessary, thereby implying that a higher LID is desirable.

---

> ### Author Response · Authors · 2024-03-18
> **Response to Reviewer U9nn part 4**
>
> **Q5:** Ablation on initial condition (base augmentation)
>
> **A5:** We employed image cropping due to its recognized importance and effectiveness in contrastive learning, as evidenced by existing studies on both natural images [4] and X-ray images [3]. Other augmentations are also plausible. We performed an additional experiment utilizing rotation as the initial augmentation choice. The experiment was carried out using the public dataset Cholec80, with downstream evaluations consistent with those outlined in the main paper.
>
> The results indicate that utilizing rotation can also surpass baseline methods. However, image cropping as base for DDA is indeed more effective compared to rotation. Please refer to the table below for the detailed results:
>
> | Initial Augmentation | Augmentation | Linear Prob (Classification) | Finetune | Finetune |
> |:--------------------:|:------------:|:-------------------------------:|:-----------:|:-----------:|
> |                      |              |          Cholec80 Tool          |   SVHM Seg  | CholecSeg8K |
> |          N/A         |    SimCLR    |              67.59              |    58.29    |    56.02    |
> |          N/A         |     Base     |              67.78              |    58.36    |    57.04    |
> |    Image Cropping    |      DDA     |            **73.59**            |  **59.31**  |  **59.40**  |
> |       Rotation       |      DDA     |              72.38              |    58.02    |    59.26    |
>
> ---
>
> [1] He, Kaiming, et al. "Momentum contrast for unsupervised visual representation learning." CVPR 2020.\
> [2] Irvin, Jeremy, et al. "Chexpert: A large chest radiograph dataset with uncertainty labels and expert comparison." AAAI 2019.\
> [3] Van der Sluijs, Rogier, et al. "Exploring image augmentations for siamese representation learning with chest x-rays." MIDL, 2023.\
> [4] Chen, Ting, et al. "A simple framework for contrastive learning of visual representations." ICML 2020.\
> [5] Tokuyasu, Tatsushi, et al. "Development of an artificial intelligence system using deep learning to indicate anatomical landmarks during laparoscopic cholecystectomy." Surgical endoscopy 35 (2021): 1651-1658.\
> [6] Silva, Bruno, et al. "Analysis of current deep learning networks for semantic segmentation of anatomical structures in laparoscopic surgery." 2022 44th Annual International Conference of the IEEE Engineering in Medicine & Biology Society (EMBC). IEEE, 2022.\
> [7] Scheikl, Paul Maria, et al. "Deep learning for semantic segmentation of organs and tissues in laparoscopic surgery." Current Directions in Biomedical Engineering. Vol. 6. No. 1. De Gruyter, 2020.\
> [8] Owen, David, et al. "Automated identification of critical structures in laparoscopic cholecystectomy." International Journal of Computer Assisted Radiology and Surgery 17.12 (2022): 2173-2181.

---

> ### Author Response · Authors · 2024-03-25
> **A follow up message**
>
> Dear reviewer U9nn,
>
> We appreciate your initial comments on our paper. In response to your review, we have conducted additional experiments and incorporated new results in our replies. Hopefully, they have addressed your concerns. Should you have any further questions or require clarification, please kindly let us know. We are more than willing to address them before the discussion concludes. We sincerely appreciate any opportunities you could provide us to improve our work.

---

### Official Review · Reviewer_cd1Y · 2024-03-06

**Confidence:** 3
**Preliminary Rating:** 3
**Final Rating:** 5

**Summary:**

This paper proposes a learnable framework for finding optimal augmentations, to be later used for self-supervised contrastive learning. This learnable framework enables to cover and test optimal set of augmentations (with their probabilities and parameters). This is done by minimizing over the augmentation set the _local intrisic dimension_ of the embedding,



(Note: this was an emergency review.)

**Strengths:**

- The problem of finding optimal augmentations is timely and relevant
- The results seem to be positive compared to pre-selected augmentation (inferred from natural images paper), and the method is reusable to other domains.
- The whole idea of using the dimensionality of the learned embedding is elegant, and very appealing to me.

**Weaknesses:**

- Overall the writing of the paper should be improved, especially in Section 2 (describing the method). See the detailed comments and questions for rebuttal, but the many small imprecision in notation do stack-up and by the end of the section many grey area remains.

**Detailed Comments:**

- Section 2.2, page 3, there is a notation clash between $\mathcal O(\cdot;p,\lambda)$ and $\bar{\mathcal O}(x;p,\lambda)$. Overall, the whole notation around $\mathbb O$, $\mathcal O$, $\bar{\mathcal O}$ is unclear, and should improved. Adding a summary of the notation in the Appendix as well would be welcome. By the time we reach Eq. (4) I am simply lost: how are you averaging over the probabilities (of the weight converted to a probability?)? Wasn't it supposed to be used inside $\mathcal O$, as described in page 3, to decide to perform the augmentation or not? The way I read Eq. (4), I understand that you are mixing different augmented images into a single image.
- In Theorem 1, $\text{LID}_F(r)$ is defined to something, which is then equated to $\text{IntrDimF}(r)$, but it is neither defined nor reused?
- The authors seem to be confusing  $f \circ g = f(g(x))$ with  $f \circ g = g(f(x))$. See [https://en.wikipedia.org/wiki/Function_composition](https://en.wikipedia.org/wiki/Function_composition). This happens at different places in the manuscript.
- Eq. (5): the variable for the minimization is not defined. (I think it is ultimately the set $\mathcal T$?) This is important as by this point a lot of notation has been introduced, and writing that explicitly would help.
- Mixing letters in iterators (as in $j = 1,...,N$) tend to make the reading more difficult. $n=1,...,N$, $k=1,...,K$ tend to be easier on the reader.

Minor:
- Write `Equation \eqref{label}` and not `equation \ref{label}` (the parenthesis of the Equations are important)

**Justification Of Final Rating:**

I thank the authors for their thorough response. Due to big changes in schedule on my end I could not participate in the discussion with the authors, I apologize for that.

I think that the manuscript has quite improved in readability, and the results and method's motivation are quite compelling.

This paper could make an interesting talk, and in any case, I would encourage the authors to explore the other applications (as mentioned in the conclusion) in a journal extension (possibly, the Melba special issue).

Good job, see you in Paris hopefully! :-)

**Justification Of The Preliminary Rating:**

Due to the imprecision in the paper, many areas are not clear to me, so I prefer to be conservative in my rating. This could be easily be upgraded (or downgraded) during rebuttal plus discussion period. I am not fully sure how to completely interpret the results either, so I'll need to clarify that.

At the same time, I am not super familiar with recent self-supervised, contrastive learning, so I could have missed parts of the existing litterature.

**Questions To Address In The Rebuttal:**

- The final framework is not clear to me. If I understood correctly, first an initial encoder $f_\theta$ is trained with only cropping used as augmentation, then the DDA algorithm is used to find an optimal set of augmentations $\mathcal T$, and then a final encoder $f_\theta$ (2?) is trained from scratch, using this time $\mathcal T$. Is this it?
- I'd like to see a discussion on the final "cost" (in time, computational ressources, other) of this proposed method and others. Notably, the LID estimator does not seem trivial either, so is it adding extra computational cost? The authors mention 4 NVidia A100 in the Appendix, which is pretty significant.
- In Algorithm 1, for $\mathcal L$ computation, an `abs` appear in the pseudo code (contrary to Eq. (5)). Why is that?
- It is not fully clear to me why the probability $p$ parameter in replaced by some weight $\boldsymbol w$, which is then put into a softmax (softmax over what? this is not clear). Cannot $p$ be learned directly?
- What is the difference between $f$ and $f_\theta$?
- What is $g$ (the projector)? Is it to be trained? Where is it defined? Where does it map to? $g: \mathcal R^d \rightarrow ?$

---

> ### Author Response · Authors · 2024-03-18
> **Response to Reviewer cd1Y part 1**
>
> Thank you very much for reviewing our paper and the valuable comments. We have prepared a clarification for each of your questions, please kindly let us know if anything is still unclear.
>
> ---
>
> **Q1:** The final framework unclear?
>
> **A1:** Yes, your understanding is correct. It's important to note that the initial encoder is essential for extracting representations used in the search process. This component is also integral to the SelfAugment method, which necessitates an additional layer on top of the initial encoder. In contrast, DDA does not require any additional model or layer.
>
> ---
>
> **Q2:** Discussion on cost (time, computational resources..)
>
> **A2:** Using our SVHM dataset as an example. For DDA, each round of pretraining takes 40 hours, the search takes 8 hours, and applying the found augmentation for final pretraining takes 40 hours, totaling 88 hours for the augmentation search and contrastive learning pretraining.
>
> If we consider the total number of possible choices of augmentation policy as $N$, using grid search would cost $N \times (40+2)$, (40 hours for pretraining and 2 hours for finetuning evaluation). For reference, the search space we used is $N=10^5$ without considering the strength parameter.
> Within our search space encompassing 10 augmentation operations, identifying a policy with 5 operations on SVHM would demand **4,200,000 hours** using grid search, whereas DDA accomplishes this task in only **48 hours**, speeding up roughly by $10^5$.
>
> Long pretraining time is necessary for SSL rather than DDA. Please refer to the SimCLR [1] paper for more details. DDA significantly speeds up the searching process to a constant 40+8 hours, rather than $N \times 42$, which depends on the search space size.
>
> The LID estimation can be performed with a single matrix multiplication to estimate the pairwise distance. It has been reported in existing work [3] that estimating LID does not introduce computation overheads. Additionally, note that LID estimation only requires use of the data within a batch, rather than use of the whole dataset.
>
>
> We have described our computational infrastructure to adhere to the MIDL reproducibility guidelines. A100 GPUs are not necessary for running DDA. There are no extra compute requirements beyond SSL pretraining. If a machine is capable of training with SSL, it can also utilize DDA.
>
>
> ---
>
> **Q3:** What is the absolute operation in the loss function in Algorithm 1?
>
> **A3:** It is not necessary to include the absolute in the implementation. We have removed the `abs` from the pseudo-code.
>
> ---
>
> **Q4:** Why the probability p  is replaced by w and need to go through a softmax, rather than being learned directly
>
> **A4:** The softmax function  $\sigma_\eta(\cdot)$ is to convert the real-number parameters $\boldsymbol{w}$ into a probability distribution. It ensures that the sum of probabilities is equal to 1. The $\boldsymbol{w}$ are learnable parameters, and $\boldsymbol{p}=\sigma_\eta(\boldsymbol{w})$. Softmax is a common procedure for  converting a real numbered value to a probability. For example, the commonly used cross-entropy loss function also uses softmax to transform logits into probabilities. We have refined the description in Section 2.2 for Equation (4).
>
> ---
>
> **Q5:** What is the difference between $f$ and $f_{\theta}$
>
> **A5:** Both terms refer to the encoder. $\theta$ is the parameters of the $f$, we omitted the $\theta$ for readability. We have updated the draft to make this simpler.
>
> ---
>
> **Q6:** What is the projector $g$?
>
> **A6:** The projector $g(\cdot)$ maps the representations $z$ to embeddings $e$. It is a multilayer perceptron (MLP) model. The contrastive learning loss is calculated with embeddings. It is trained during the pretraining. It will be disregarded after pretraining. This is the standard model architecture for SSL [1, 4, 5]. There is an excellent illustration here, a published open-source project for SimCLR: [https://github.com/google-research/simclr](https://github.com/google-research/simclr). In the Figure, CNN is the encoder, and MLP is the projector.
>
> ---

---

> ### Author Response · Authors · 2024-03-18
> **Response to Reviewer cd1Y part 2**
>
> **Q7:** Notation for Section 2.2
>
> **A7:** There is no clash in $\bar{\mathcal{O}}$, $\mathcal{O}$ and $\mathbb{O}$. The $\bar{\mathcal{O}}$ is an augmentation policy, which is the result of applying the augmentation operation $\mathcal{O}$ with probability $p$ (at the bottom of page 3). The $\mathbb{O}$ denotes the set of all possible augmentation operations. We follow existing works [6, 7] that use these notations. The definition at the bottom of page 3 is a simplified case with the augmentation operation $\mathcal{O}_k$ being applied with $p_k$, otherwise the other operations from $\mathbb{O}$.
>
>
> For the search, we are interested in multiple different augmentations. Eq (4) is the mixing of different augmented images with their corresponding probability into a single output used during the differentiable search. Eq (4) is an approximation of selecting multiple augmentation operations. Each operation $\mathcal{O}_k$ with their probability $p_k$ and strength $\lambda_k$, the final output is $\bar{\mathcal{O}}$, which is defined as their weighted sum (the Eq(4)). This is a standard approximation technique in the differentiable search frameworks [7, 8].
>
> During the SSL training, we sample an augmentation operation per layer according to the optimized parameter (using softmax to convert to probability). In other words, the Eq (4) is only used during the search.
>
> In the revision, we have updated the draft to make this clearer.
>
>
> ---
>
>
> **Q8:** Theorem 1 and LID definition
>
> **A8:** The $\mathrm{IntrDimF(r)}$ refers to the general definition of LID [9], and it applies to the more general cases. In our case, following existing work [3], we are only interested in a function $F$ that satisfies the conditions of a cumulative distribution function(CDF), which is Theorem 1. We have removed the $\mathrm{IntrDimF(r)}$ in the revision to avoid confusion.
>
>
> ---
> **Q9:** Notation for function composition, Eq (5), iterators, and Equation
>
> **A9:** We have updated these typos in the updated revision.
>
> ---
> [1] Chen, Ting, et al. "A simple framework for contrastive learning of visual representations." ICML 2020.\
> [2] Van der Sluijs, Rogier, et al. "Exploring image augmentations for siamese representation learning with chest x-rays." MIDL, 2023.\
> [3] Huang, Hanxun, et al. "LDReg: Local Dimensionality Regularized Self-Supervised Learning." ICLR 2024.\
> [4] Grill, Jean-Bastien, et al. "Bootstrap your own latent-a new approach to self-supervised learning." NeurIPS 2020.\
> [5] Bardes, Adrien, Jean Ponce, and Yann LeCun. "VICReg: Variance-Invariance-Covariance Regularization for Self-Supervised Learning." ICLR 2021.\
> [6] Lim, Sungbin, et al. "Fast autoaugment." NeurIPS (2019).\
> [7] Hataya, Ryuichiro, et al. "Faster autoaugment: Learning augmentation strategies using backpropagation." ECCV, 2020.\
> [8] Liu, Hanxiao, Karen Simonyan, and Yiming Yang. "DARTS: Differentiable Architecture Search." ICLR. 2019.\
> [9] Houle, Michael E. "Local intrinsic dimensionality I: an extreme-value-theoretic foundation for similarity applications." SISAP 2017.

---

> ### Author Response · Authors · 2024-03-25
> **Any additional questions?**
>
> Dear reviewer cd1Y,
>
> Thanks again for reviewing our paper. Please kindly let us know if you have any additional questions or require further clarification. We are happy to address them before the discussion ends.

---

### Author Response · Authors · 2024-03-18
**Summary of changes in the revision**

We appreciate the insightful reviews we have received and have implemented the following changes to the draft based on reviewers' suggestions. These modifications are also reflected in the responses to the reviewers below.

- Results of supervised learning (without pretraining) were added to Table 1, to address concerns raised by wQKj. It clearly shows the benefits of using SSL to enhance performance.
- Results of manually selected augmentations, based on literature settings [3, 4, 5, 6], were added to Table 1 as a baseline method denoted as *Manual*. This is to address reviewer U9nn’s concern that the baseline should include a set of reasonable augmentations known in the literature. It can be observed that augmentations effective in supervised segmentation for laparoscopic images do not perform well in contrastive learning.
- To address common concerns raised by reviewers U9nn and wQKj, an ablation study on the augmentation choice for the initial encoder was added in Appendix B.5. It is observed that the choice of basic augmentation does not significantly affect the effectiveness of DDA.
- In Appendix B.6, an experiment was conducted performing the search with an X-ray dataset (CheXpert [7]) and a natural image dataset (CIFAR10 [8]). Results indicate that DDA can find appropriate augmentations. This change is to address the concerns of reviewer U9nn. Although the final performance is marginally lower than with grid search (exhaustive search over all possible combinations), DDA can identify an augmentation policy that closely matches the best one for each dataset without requiring supervised evaluation. Moreover, DDA has considerably lower computational complexity than grid search.
- The augmentation policy found by SelfAugment for the private SVHM dataset was added into Table 5, and those for Cholec80 presented in Table 6, as requested by reviewer wQKj.
- We provided the visualization of a randomly selected batch of images from the SVHM dataset, showcasing the application of various augmentation policies. These visualizations are available in Figure 12 through 16. This is to address the concern by reviewer wQKj.
- Captions and axis titles were added to the heatmaps in Figure 6 through 10. This is to address the concern by reviewer wQKj.
- We updated the typos and suggested notations from the reviewer cd1Y and U9nn.

Finally, we wish to emphasize the efficiency of DDA, distinguishing it from existing methods. Traditional approaches, such as grid search, as employed in studies concerning natural images [1] and X-ray datasets [2], require exhaustive exploration of all possible combinations alongside pretraining. Moreover, selecting the optimal policy entails finetuning the pretrained model on labeled datasets. This process escalates exponentially in duration based on the search space and the number of operations in the policy.

In contrast, DDA requires only one pretraining session alongside the search, without any prerequisites for labeled dataset evaluation. This results in constant searching time, regardless of the number of potential combinations. To illustrate, within our search space encompassing 10 augmentation operations, identifying a policy with 5 operations on SVHM would demand roughly **4,200,000 hours** using grid search, whereas DDA accomplishes this task in only **48 hours**, speeding up by $10^5$. This underscores DDA's suitability for diverse medical datasets with unique characteristics, as it circumvents the computationally exhaustive nature of grid search.

---

[1] He, Kaiming, et al. "Momentum contrast for unsupervised visual representation learning." CVPR 2020.\
[2] Van der Sluijs, Rogier, et al. "Exploring image augmentations for siamese representation learning with chest x-rays." MIDL, 2023.\
[3] Tokuyasu, Tatsushi, et al. "Development of an artificial intelligence system using deep learning to indicate anatomical landmarks during laparoscopic cholecystectomy." Surgical endoscopy 35 (2021): 1651-1658.\
[4] Silva, Bruno, et al. "Analysis of current deep learning networks for semantic segmentation of anatomical structures in laparoscopic surgery." 2022 44th Annual International Conference of the IEEE Engineering in Medicine & Biology Society (EMBC). IEEE, 2022.\
[5] Scheikl, Paul Maria, et al. "Deep learning for semantic segmentation of organs and tissues in laparoscopic surgery." Current Directions in Biomedical Engineering. Vol. 6. No. 1. De Gruyter, 2020.\
[6] Owen, David, et al. "Automated identification of critical structures in laparoscopic cholecystectomy." International Journal of Computer Assisted Radiology and Surgery 17.12 (2022): 2173-2181.\
[7] Irvin, Jeremy, et al. "Chexpert: A large chest radiograph dataset with uncertainty labels and expert comparison." AAAI 2019.\
[8] Krizhevsky, Alex, and Geoffrey Hinton. "Learning multiple layers of features from tiny images." 2009.

---

### Author Response · Authors · 2024-03-26
**Keen request for discussion**

Dear AC,

We have provided a point-to-point response to each reviewer's concerns. We have yet to receive any feedback from reviewers in the discussion period. We would appreciate it if you could remind reviewers to participate in the discussion.

Thank you for your attention to this matter.


Authors.

---

### Meta-Review · Area_Chair_EXp5 · 2024-04-02

**Recommendation:** Accept (Poster)
**Confidence:** 5

**Metareview:**

After revision, there is clear reviewer agreement (reviewer wQKj updated through private channels) that the strengths of the manuscript now clearly outweigh the weaknesses and that the paper can be accepted at MIDL.

However, the response to reviewers included several additional experiments and results, and so it is mandatory that these new findings be included in the final manuscript.

---

### Decision · Program_Chairs · 2024-04-06

Accept (Poster)